# Bayesian Optimization of Function Networks

**Raul Astudillo**
Cornell University
ra598@cornell.edu

**Peter I. Frazier**
Cornell University
pf98@cornell.edu

## Abstract

We consider Bayesian optimization of the output of a network of functions, where each function takes as input the output of its parent nodes, and where the network takes significant time to evaluate. Such problems arise, for example, in reinforcement learning, engineering design, and manufacturing. While the standard Bayesian optimization approach observes only the final output, our approach delivers greater query efficiency by leveraging information that the former ignores: intermediate output within the network. This is achieved by modeling the nodes of the network using Gaussian processes and choosing the points to evaluate using, as our acquisition function, the expected improvement computed with respect to the implied posterior on the objective. Although the non-Gaussian nature of this posterior prevents computing our acquisition function in closed form, we show that it can be efficiently maximized via sample average approximation. In addition, we prove that our method is asymptotically consistent, meaning that it finds a globally optimal solution as the number of evaluations grows to infinity, thus generalizing previously known convergence results for the expected improvement. Notably, this holds even though our method might not evaluate the domain densely, instead leveraging problem structure to leave regions unexplored. Finally, we show that our approach dramatically outperforms standard Bayesian optimization methods in several synthetic and real-world problems.

## 1 Introduction

We consider Bayesian optimization (BO) of objective functions defined by a series of time-consuming-to-evaluate functions, $f_1, \ldots, f_K$, arranged in a directed acyclic network, so that each function takes as input the output of its parent nodes. As we detail below, these problems arise in BO-based policy search in reinforcement learning (Lizotte et al., 2007), optimization of complex systems modeled via simulation, and calibration of time-consuming physics-based models.

To illustrate, we introduce a running example of vaccine manufacturing (Sekhon and Saluja, 2011), focusing on the portion of the manufacturing process that uses live cells to produce proteins needed in a vaccine. It begins with a cell culture, in which living cells are grown and used as "factories" to produce proteins. This process is controlled by a vector, $x_1$, containing the temperature, pH, and $CO_2$ content used when growing these cells. The output of this process is the quantity of the desired protein $y_1 = f_1(x_1)$, i.e., the *yield* of this step, along with other byproducts. This output is passed into a second process, purification, which removes byproducts and is controlled by a vector $x_2$ comprising temperature, pressure, and flow rate. The yield of the desired protein from this second step is $y_2 = f_2(x_2, y_1)$. This output enters a third step, formulation, in which we formulate the raw protein into a form that can be distributed as controlled by a third set of parameters. This determines the yield of the overall process $y_3 = f_3(x_3, y_2)$. We wish to choose $(x_1, x_2, x_3)$ to maximize overall protein yield. This problem is summarized as a function network in Figure 1.

The problem described above and other similar problems can be tackled with Bayesian optimization (BO), which has been shown to perform well compared to other derivative-free global optimization

35th Conference on Neural Information Processing Systems (NeurIPS 2021).

methods for time-consuming-to-evaluate objective functions (Snoek et al., 2012; Frazier, 2018). A standard BO algorithm would fit a Gaussian process (GP) (Rasmussen and Williams, 2006) model on the objective function ($y_3$, which depends on $(x_1, x_2, x_3)$) and use it, along with an acquisition function, to sequentially choose the points to evaluate. Under this standard approach, however, evaluations of the intermediate nodes, $f_1, \ldots, f_{K-1}$, would be ignored despite being available when computing the objective function. In the example above, this corresponds to looking only at the yield of the overall process, and not of each individual step.

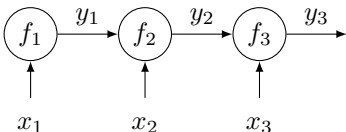

Figure 1: Vaccine manufacturing as a function network. Protein $y_1 = f_1(x_1)$ is created, then purified with yield $y_2 = f_2(x_2, y_1)$, and formulated with yield $y_3 = f_3(x_3, y_2)$. The goal is to find $(x_1, x_2, x_3)$ that maximizes $y_3$.

In this paper, we introduce a novel BO approach that leverages function network structure for substantially more efficient optimization. This approach models the individual nodes of the network using distinct GPs. This allows incorporating observations of each node's output recursively into a non-Gaussian posterior on the network's overall output. Our approach then chooses the points to evaluate using the expected improvement (Jones et al., 1998) computed with respect to this implied posterior on the objective function. The non-Gaussian nature of this posterior prevents the expected improvement from having a closed form. However, we show that it can still be efficiently maximized via sample average approximation (Kleywegt et al., 2002).

Our approach can outperform standard BO by leveraging information from internal nodes unavailable to standard methods. We briefly explain one way this can happen, in the context of the example above. In vaccine manufacturing, each function $f_k(x_k, y_{k-1})$, $k = 2, 3$, is bounded between 0 and $y_{k-1}$ because new protein cannot be created in the purification and formulation stages. Moreover, application experts have a prior on what values for $f_k(x_k, y_{k-1})/y_{k-1}$ should be achievable if $x_k$ is set well. Thus, if we see that $y_3$ is unexpectedly poor, information from intermediate nodes can be extremely valuable: if $y_1$ and $y_2$ are as expected, then this suggests the problem is with $x_3$; if $y_1$ is as expected but $y_2$ is poor, then the problem is likely with $x_2$; and if $y_1$ is poor then the problem is likely with $x_1$. (If there is a problem with $x_k$, there may also be a problem with $x_{k'}$, $k' > k$, but we can focus on fixing $x_k$ first.) Thus, by observing intermediate nodes, we can instantly reduce the effective dimensionality of the input space by a factor of $K = 3$.

We show that our method is asymptotically consistent, i.e., that it discovers the global optimum given sufficiently many samples. Remarkably, in contrast with most BO methods, it may do so without measuring densely over the feasible domain, instead leveraging function network structure to exclude regions as unnecessary to explore. This indicates the power of function network structure to improve query efficiency.

We demonstrate through numerical experiments that access to additional information available in a problem formulated as a function network can dramatically accelerate optimization. We study four synthetic problems and four real-world problems: a manufacturing problem similar in spirit to the vaccine example above, an active learning problem with a robotic arm, and two problems arising in epidemiology, one calibrating an epidemic model and the other designing a testing strategy to control the spread of COVID-19. Our method significantly outperforms competing methods that utilize less information, in some cases by $\sim 5\%$ and in other cases by several orders of magnitude.

## 2   Related Work

Our work occurs within BO, a framework for global optimization of expensive-to-evaluate black-box functions that originated with the work of Zhilinskas (1975) and Močkus (1975), and has recently become popular due to its remarkable performance in hyperparameter tuning of machine learning algorithms (Snoek et al., 2012; Swersky et al., 2013; Wu et al., 2019). We refer the reader to Shahriari et al. (2016) and Frazier (2018) for modern introductions to BO.

Our approach can be catalogued as a *grey-box* BO method since it does not treat the objective function entirely as a black box, and instead exploits known structure to improve sampling efficiency. Other examples of grey-box BO approaches include multi-fidelity BO (Kandasamy et al., 2017; Wu et al., 2019), which leverages cheaper approximations of the objective function; BO of objective functions that are the integral of an expensive-to-evaluate integrand (Williams et al., 2000; Toscano-Palmerin and Frazier, 2018; Cakmak et al., 2020); BO of objective functions that are a sum of squared errors (Uhrenholt and Jensen, 2019); and, more generally, BO of objective functions that are the composition of an expensive-to-evaluate inner function and a known inexpensive-to-evaluate outer function (Astudillo and Frazier, 2019), of which our work can be seen as a significant generalization. We refer the reader to Astudillo and Frazier (2021) for a tutorial on grey-box BO.

Our work is also closely related to Marque-Pucheu et al. (2019), which proposes a method for efficient sequential experimental design of nested computer codes, also using GPs. This work focuses on the case where there are only two node functions, and one takes as input the output of the other. In contrast with our work, the goal of the proposed method is to learn the output code as accurately as possible within a limited evaluation budget, but optimization is not pursued.

Optimization of composite (a.k.a nested) functions has also been considered in the gradient-based optimization literature (Shapiro, 2003; Drusvyatskiy and Paquette, 2019; Charisopoulos et al., 2021; Balasubramanian et al., 2020). In contrast with ours, these works assume that evaluations are inexpensive, and often also some form of convexity, along with availability of gradients. However, this literature is similar in spirit to ours in that information from inner functions improves efficiency.

Function networks arise in many application areas. While these applications have not, to our knowledge, been previously formulated as specific instances of the general function network model we propose, their literatures are nonetheless relevant to our work.

- In engineering and aerospace design, function networks arise in multidisciplinary optimization (Cramer et al., 1994; Amaral et al., 2014; Benaouali and Kachel, 2019), where simulators focusing on different physical laws are coupled into a function network. For example, a simulation of airflow over a wing may output the forces on the wing to another simulation of mechanical stress on the wing structure.

- In drug discovery and materials design, function networks arise from the sequence-structure-function (Sadowski and Jones, 2009) and composition-structure-property (Hattrick-Simpers et al., 2016) paradigms. Here, decision variables (composition, e.g., what fraction of what raw materials are used) determine the structure of the material (how the atoms combine together), which in turn determines how the material behaves (properties).

- Function networks arise in the design of queuing networks (Fu and Henderson, 2017; Arcelli, 2020). This includes manufacturing systems (Ghasemi et al., 2018), where the partially-processed output of one workstation is input to the next workstation, the design of service systems (Wang et al., 2020) such as hospitals and airport security checkpoints, and choosing traffic signal timings for a city's street network (Osorio and Bierlaire, 2013).

- Finally, function networks arise in reinforcement learning (Sutton and Barto, 2018) and Markov decision processes (Puterman, 1990), where the transition kernel transforms the state variable at the start of a time range to another state variable at the end of this range. This outputted state variable is the input to the next time range. §5.2 shows an example.

## 3   Problem Setting

We consider objective functions evaluated via a series of functions, $f_1, \ldots, f_K$, arranged in a directed network so that each function in this network takes as input the output of its parent nodes, and assume that evaluating this collection of functions is time-consuming. The network structure is encoded as a directed graph with nodes $V = \{1, \ldots, K\}$ and directed edges $E = \{(j, k) : f_k$ take as input the output of $f_j\}$. We assume that this graph is acyclic and has a single leaf node whose output is the objective to optimize.

Let $J(k)$ denote the set of parent nodes of node $k$[1]. We assume, without loss of generality, that the nodes are ordered such that $j < k$ for all $j \in J(k)$. In addition to the output of its parent nodes, we

---

[1]We allow $J(k) = \emptyset$, which corresponds to the root node(s).

assume that each function, $f_k$, takes as input a (potentially empty) subset, $I(k) \subset \{1, \ldots, D\}$, of the components of the decision vector $x \in \mathbb{R}^D$.

Let $h_1(x), \ldots, h_K(x)$ denote the values of the $K$ nodes in the function network when it is evaluated at $x$. These values are defined recursively by

$$h_k(x) = f_k\left(x_{I(k)}, h_{J(k)}(x)\right), \ k = 1, \ldots, K,$$

i.e., by evaluating each function in the network as the values of its parent nodes become available. This recursion is well-defined by our assumption that the graph is acyclic. The objective function is then $g = h_K$, and the goal is to solve

$$\max_{x \in \mathbb{X}} g(x),$$

where $\mathbb{X} \subset \mathbb{R}^D$ is a simple compact set, such as a hyper-rectangle.

The standard BO approach to this problem models $g$ using a GP prior distribution. This approach iteratively chooses the next point at which to evaluate $g$ as follows. Given $n$ observations of the objective function, $g(x_1), \ldots, g(x_n)$, it computes the posterior distribution on $g$, which is itself a GP. It then uses this posterior distribution to compute an acquisition function (Frazier, 2018) that quantifies the value of the information that would result from a function evaluation at any given point. Finally, it chooses the next point to evaluate, $x_{n+1}$, as the point that maximizes this acquisition function. Notably, although the outputs of $f_1, \ldots, f_{K-1}$ would be observed as part of this approach, these evaluations would be ignored by standard BO when calculating the posterior distribution and corresponding acquisition function.

## 4 Bayesian Optimization with Full Network Observations

This section describes our approach. Like standard BO, it consists of a statistical model and an acquisition function. Unlike standard BO, however, our approach leverages the network structure of the problem by utilizing intermediate outputs within the network. As we describe below, this is achieved by modeling the node functions, $f_1, \ldots, f_K$, as GPs, which in turn implies a non-Gaussian distribution on $g$ (§4.1). Our acquisition function is the expected improvement under this posterior distribution (§4.2), which no longer has a closed form and thus we maximize it via sample average approximation (§4.3). We end up this section by proving that our acquisition function is asymptotically consistent despite not necessarily sampling $\mathbb{X}$ densely (§4.4).

### 4.1 Statistical Model

Instead of modeling $g$ directly, we model $f_1, \ldots, f_K$, as drawn from independent GP prior distributions. Let $\mu_{0,k}$ and $\Sigma_{0,k}$ denote the prior mean and covariance functions of $f_k$, $k = 1, \ldots, K$, respectively. When $g$ is evaluated at $x$, we also get to observe the value of $f_k$ at $\left(x_{I(k)}, h_{J(k)}(x)\right)$. Thus, after querying $g$ at $n$ points, $x_\ell$, $\ell = 1, \ldots, n$, the observations of the values of $f_k$, $k = 1, \ldots, K$, at $\left(x_{\ell,I(k)}, h_{J(k)}(x_\ell)\right)$, $\ell = 1, \ldots, n$, imply posterior distributions on $f_1, \ldots, f_K$, which are again (conditionally) independent GPs with mean and covariance functions $\mu_{n,k}$ and $\Sigma_{n,k}$[2], $k = 1, \ldots, K$. These can be computed in closed form using the standard GP regression equations (see, e.g., Rasmussen and Williams 2006). For completeness, we include these equations in §F of the supplement.

The posterior distributions on $f_1, \ldots, f_K$, described above imply a posterior distribution on $g$. This distribution is in general non-Gaussian. Thus, unlike in the standard setting, where $g$ is modeled as a GP, classical acquisition functions such as the expected improvement cannot be computed in closed form. However, as we describe next, drawing samples from this distribution is simple.

Thanks to the acyclic structure of the underlying network that defines $g$, a sample, $\widehat{g}(x) = \widehat{h}_K(x)$ from the posterior distribution on $g$ at $x$ can be obtained following the iterative process described next. In each iteration, $k = 1, \ldots, K$, we obtain a sample, $\widehat{h}_k(x)$, from the posterior distribution on $h_k(x)$ by drawing a sample from the normal distribution with mean $\mu_{n,k}\left(x_{I(k)}, \widehat{h}_{J(k)}(x)\right)$ and

---

[2]For ease of presentation we assume that all the functions in the network are expensive-to-evaluate, and thus require to be modeled as GPs. However, if any of the functions, say $f_k$, is not expensive-to-evaluate, we can simply take $\mu_k \equiv f_k$ and $\Sigma_k \equiv 0$.

standard deviation

$$\sigma_{n,k}\left(x_{I(k)}, \widehat{h}_{J(k)}\right) = \Sigma_{n,k}\left(x_{I(k)}, \widehat{h}_{J(k)}(x), x_{I(k)}, \widehat{h}_{J(k)}(x)\right)^{1/2}.$$

Supporting efficient calculation, the samples $\widehat{h}_{J(k)}(x)$ will have already been obtained in previous iterations of the for loop since $j < k$ for all $j \in J(k)$ (Note that $J(1) = \emptyset$). This procedure is summarized in Algorithm 1.

---

**Algorithm 1** Draw one sample from the posterior on $g(x)$

---

**Require:** $x \in \mathbb{X}$
1: **for** $k = 1, \dots, K$ **do**
2:     $\widehat{h}_k(x) \sim \mathcal{N}\left(\mu_{n,k}\left(x_{I(k)}, \widehat{h}_{J(k)}(x)\right), \sigma_{n,k}\left(x_{I(k)}, \widehat{h}_{J(k)}(x)\right)^2\right)$
3: **end for**
4: **return** $\widehat{g}(x) = \widehat{h}_K(x)$

---

We end this section by noting that, while the statistical model described above could be catalogued as a deep GP (Damianou and Lawrence, 2013), in the sense that we have GP layers in an architecture described by a directed acyclic graph, inference in our model is faster. Typically, deep GP inference requires marginalization over latent values of GP layers. In our setting, however, observation structure creates conditional independence across layers, avoiding the need to marginalize.

## 4.2 Expected Improvement for Function Networks

Our acquisition function is the expected improvement, computed with respect to the implied posterior distribution on $g$ under the statistical model described in §4.1:

$$\text{EI-FN}_n(x) = \mathbb{E}_n\left[\{g(x) - g_n^*\}^+\right],$$

where $g_n^* = \max_{i=1,\dots,n} g(x_n)$ is the best value observed so far, and $\mathbb{E}_n$ is the expectation computed with respect to the GP time-$n$ posterior distributions on $f_1, \dots, f_K$. To distinguish it from the classical expected improvement, we refer to our acquisition function as the expected improvement for function networks (EI-FN) in the remainder of this work.

## 4.3 Maximization of EI-FN via Sample Average Approximation

Since the posterior distribution on $g$ is non-Gaussian, in contrast with the classical expected improvement acquisition function, $\text{EI-FN}_n$ does not admit a closed form expression. However, $\text{EI-FN}_n(x)$ can be computed with arbitrary precision following a simple Monte Carlo (MC) approach:

$$\text{EI-FN}_n(x) \approx \frac{1}{M}\sum_{m=1}^{M}\left\{\widehat{g}(x)^{(m)} - g_n^*\right\}^+,$$

where $\widehat{g}(x)^{(1)}, \dots, \widehat{g}(x)^{(M)}$ are samples drawn from the posterior distribution on $g(x)$, which can be obtained following the approach described in §4.1.

Following the above MC scheme to approximately compute EI-FN, one can aim to maximize this function using a derivative-free global optimization algorithm for inexpensive-to-evaluate functions, such as CMA (Hansen, 2016), by drawing samples, $\widehat{g}(x)^{(1)}, \dots, \widehat{g}(x)^{(M)}$ independently for each $x$ at which $\text{EI-FN}_n$ is evaluated. However, this approach is slow since evaluations of EI-FN are noisy and derivative information is not leveraged. Instead, we propose to maximize EI-FN following a sample average approximation (SAA) approach (Kleywegt et al., 2002; Balandat et al., 2020).

Succinctly, the SAA approach works by building a MC approximation of EI-FN that is deterministic given a finite set of random variables not depending on $x$, and maximizing this approximation instead of EI-FN itself. The key observation for creating this approximation is that a sample, $\widehat{g}(x)$, can be obtained as $\widehat{g}(x) = \widehat{h}_K(x)$, where $\widehat{h}_1(x), \dots, \widehat{h}_K(x)$ are defined recursively by

$$\widehat{h}_k(x; Z) = \mu_{n,k}\left(x_{I(k)}, \widehat{h}_{J(k)}(x; Z)\right) + \sigma_{n,k}\left(x_{I(k)}, \widehat{h}_{J(k)}(x; Z)\right) Z_k,$$

where $Z = (Z_1, \ldots, Z_K)^\top$ is a standard normal random vector, and we write $\widehat{h}_k(x)$ as $\widehat{h}_k(x; Z)$ to make its dependence on $Z$ explicit. Analogously, we also write $\widehat{g}(x)$ as $\widehat{g}(x; Z)$. This can be seen as an extension of the so-called reparametrization trick for acquisition functions (Wilson et al., 2018).

If we now fix $M$ samples drawn from the $K$-dimensional standard normal distribution, $Z^{(1)}, \ldots, Z^{(M)}$, then

$$\widehat{\text{EI-FN}}_n \left( x; Z^{(1:M)} \right) := \frac{1}{M} \sum_{m=1}^{M} \left\{ \widehat{g} \left( x; Z^{(m)} \right) - g_n^* \right\}^+$$

is a MC approximation of EI-FN that is deterministic given $Z^{(1)}, \ldots, Z^{(M)}$, as desired. Moreover, Proposition 1 below shows that, under mild regularity conditions, any maximizer of the above SAA converges in probability exponentially fast to a maximizer of EI-FN$_n$ as $M \to \infty$, thus suggesting that in practice it is safe to use small values of $M$. This result is a generalization of Theorem 1 in Balandat et al. (2020). Its proof can be found in §A of the supplement.

**Proposition 1.** *Suppose that the functions $\mu_{n,k}$ and $\sigma_{n,k}$, $k = 1, \ldots, K$, are all Lipschitz continuous and let*

$$\widehat{x}_*^{(M)} \in \operatorname*{argmax}_{x \in \mathbb{X}} \widehat{\text{EI-FN}}_n \left( x; Z^{(1:M)} \right), \quad X_* = \operatorname*{argmax}_{x \in \mathbb{X}} \text{EI-FN}_n(x).$$

*Then, for every $\epsilon > 0$, there exist $A, \alpha > 0$ such that $\mathbb{P} \left( \operatorname{dist} \left( \widehat{x}_*^{(M)}, X_* \right) > \epsilon \right) \leq A e^{-\alpha M}$, $M \in \mathbb{N}$.*

Finally, we note that $\widehat{\text{EI-FN}}_n$ is differentiable with respect to $x$, provided that $\mu_{k,n}$ and $\Sigma_{k,n}$, $k = 1, \ldots, K$, are all differentiable. Thus, $\widehat{\text{EI-FN}}_n$ can be maximized using a gradient-based deterministic optimization algorithm such as L-BFGS-B (Byrd et al., 1995), with multiple restarts.

### 4.4 Asymptotic Consistency of EI-FN without Dense Measurements

To shed light on the convergence properties of the expected improvement acquisition function under our statistical model, we prove that, under suitable regularity conditions, EI-FN is asymptotically consistent, meaning that it finds the global optimum of the objective function as the number of evaluations grows to infinity. This builds on results shown for the classical expected improvement (Vazquez and Bect, 2010; Bect et al., 2019) and the expected improvement for composite functions (Astudillo and Frazier, 2019), and we rely on similar assumptions. This result is stated in Theorem 1 below and its proof can be found in §B of the supplement.

**Theorem 1.** *Suppose that $x_{n+1} \in \operatorname{argmax}_{x \in \mathbb{X}} \text{EI-FN}_n(x)$ for all $n$. Then, under regularity conditions stated precisely in the supplement, $g_n^* \to \max_{x \in \mathbb{X}} g(x)$ as $n \to \infty$ almost surely under the prior distribution on $f_1, \ldots, f_K$.*

Significant novelty in our proof arises from the fact that EI-FN's measurements are not necessarily dense in $\mathbb{X}$. In nearly all existing work, consistency of BO methods is shown by first showing that the measurements are dense in the objective's domain (see, e.g., Vazquez and Bect 2010). Surprisingly, the function network structure may allows us to exclude certain regions of $\mathbb{X}$ as suboptimal after only finitely many measurements, allowing our method to be consistent without measuring everywhere. Such ability gives insight into EI-FN's strong empirical performance compared to methods ignoring function network structure. As stated in Proposition 2 below, we provide a function network where EI-FN provides a consistent estimate of the global optimum *without* sampling densely.

**Proposition 2.** *There exists a function network (detailed in §C of the supplement) in which EI-FN is consistent but whose measurements are not necessarily dense in $\mathbb{X}$.*

## 5 Numerical Experiments

We compare the performance of our algorithm (EI-FN) against the classical expected improvement (EI), i.e., the expected improvement under a GP model over $g$. We also compare with the performance of two other standard algorithms: the algorithm that chooses the points to evaluate uniformly at random over $\mathbb{X}$ (Random); and the knowledge gradient (KG) (also under a GP model over $g$), a more sophisticated acquisition function that often delivers a better performance than the expected improvement (Wu and Frazier, 2016). The problems in §5.2 and §5.3 fall within the framework

of Astudillo and Frazier (2019) and we include its proposed EI-CF algorithm as a benchmark. All algorithms were implemented in BoTorch (Balandat et al., 2020).

We evaluate on 4 synthetic problems and 5 real-world problems. These are described below or in §D of the supplement, with the supplement describing a manufacturing problem building on §1, a COVID-19 testing problem building on §5.3, and a robot control problem similar to the one described in §5.2. In all problems, a first stage of evaluations is performed using $2(d + 1)$ points chosen uniformly at random over $\mathbb{X}$. A second stage (pictured in plots) is then performed using each of the algorithms. Error bars in Figure 2 show the mean of the best objective value observed so far, plus and minus 1.96 times the standard deviation divided by the square root of the number of replications. Since the difference in performance in some of our experiments is better appreciated in a logarithmic scale, we also include plots showing the $\log_{10}$-regret for such experiments in Figure 3. Each experiment was replicated 30 times. Experimental setup details and runtimes are available in the supplement. Code to reproduce our numerical experiments can be found at `https://github.com/RaulAstudillo06/BOFN`.

## 5.1 Synthetic Test Functions

We create synthetic test problems by arranging standard test functions from the global optimization literature (Jamil and Yang, 2013) into function networks. These explore a variety of network structures in an easy-to-reproduce form, and are named after the standard test function used to define the function network. We describe these briefly here and then in full detail in §D of the supplement.

**Alpine2** and **Rosenbrock** both arrange $K$ nodes in series, where each node except the first node takes the output of the previous node as input. Additionally, in Alpine2, each node takes a distinct dimension of the decision vector $x$ as input. In Rosenbrock, $x_1$ and $x_2$ are inputs to the first node, $x_2$ and $x_3$ are inputs to the second, and so on. These network architectures arise in manufacturing problems like the example in §1, as well as business operations with queues like boarding an aircraft or fulfilling drive-through orders. For Alpine2 we set $K = 6$, and for Rosenbrock we set $K = 4$.

**Ackley** has 3 nodes. The first two nodes each take the same 6-dimensional input. Their outputs are passed to the third node that produces the final output. This type of function network arises in algorithm design for two-sided markets (Li et al., 2021), like Uber and AirBnB, where the first node simulates an intervention's effect on riders (or guests), the second simulates its effect on drivers (or hosts), and the third simulates the matching process where riders and drivers (or guests and hosts) interact to produce transactions.

**Drop-Wave** has two nodes. The first node takes a two-dimensional vector $x$ as input. This node's output is passed to the second node, which produces the objective value. This network architecture is representative of multidisciplinary engineering design (Benaouali and Kachel, 2019), for example in aerospace, where a small number of distinct black-box simulators simulate processes governed by physical laws that affect each other through a small number of channels, such as an aircraft engine simulation (the first node) determining heat generated while flying, which is then inputted to a temperature-dependent simulation of mechanical stress on the aircraft's frame (the second node).

## 5.2 Fetch-and-Reach with a Robotic Arm

This test problem is obtained by adapting the Fetch environment from OpenAI Gym (see Plappert et al. (2018)). The goal is to move the gripper of a robotic arm to a target location with only three movements. We formulated this problem as a function network with 3 nodes, each representing a movement of the robotic arm. Each of these nodes takes as input the current location of the gripper along with a vector of forces to be applied to the robotic arm in that step, and produces as output the location of the arm after this movement is complete. (Note that the output of each node is 3-dimensional and thus this can also be thought of as a function network with 9 single-output nodes). The objective to minimize is the distance between the gripper and the target in the final step. Figure 4 shows an animation of this problem.

We formalize the above problem as follows. Let $z_{\mathrm{init}}, z_{\mathrm{target}} \in \mathbb{R}^3$ denote the object's initial and target locations, respectively. At each time step, $t$, we choose the vector of forces to be applied to the robotic arm $x_t \in [-1, 1]^3$. After this movement, the location of the object becomes $z_{t+1}$. The goal is to choose $x_t$ for $t = 1, \ldots, T$ to minimize $\|z_{\mathrm{target}} - z_T\|_2$. We set $z_{\mathrm{init}} = (0, 0, 0)$,

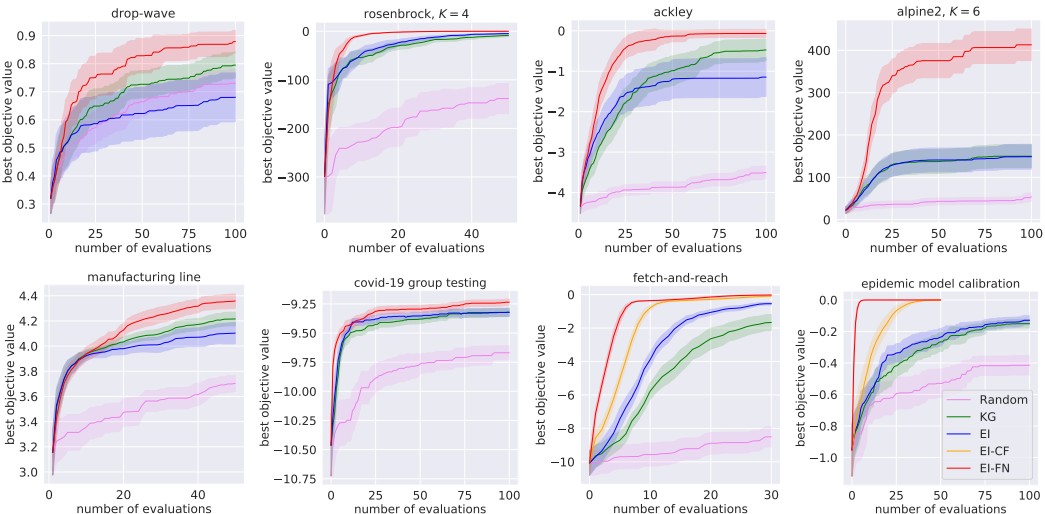

Figure 2: Top: Results on synthetic problems that adapt widely used synthetic test functions into function networks. Bottom: Results on realistic problems: manufacturing line, design of testing protocols for COVID-19, fetch-and-reach with a robotic arm, and calibration of an epidemic model. EI-FN substantially improves over benchmark methods, with larger improvements for problems with higher-dimensional decision vectors and more nodes.

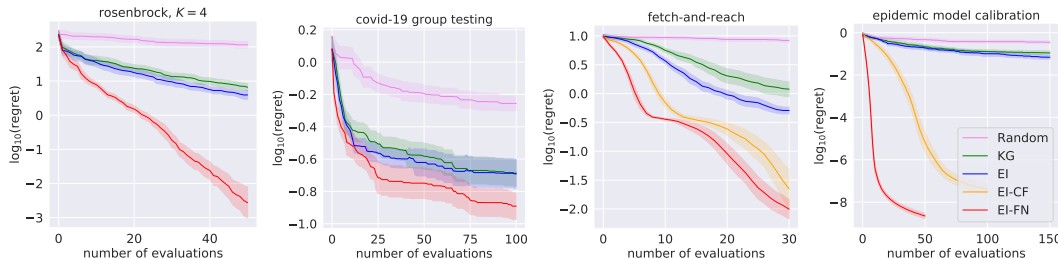

Figure 3: Results on four of our test problems. In contrast with Figure 2 above, which shows the best objective value found, here we plot the $\log_{10}$-regret.

$z_{\text{target}} = (-12, 13, 0.2)$, and $T = 3$. This can be interpreted as a function network by associating each time step with a triplet of node functions $f_t = (f_{t,1}, f_{t,2}, f_{t,3})$ which take $x_t$ as input and produce $x_{t+1} = f_t(x_t)$ as output.

A very similar experiment to the one described above can be found in §E of the supplement. It considers a a variation of the active learning for robot pushing problem introduced by Wang and Jegelka (2017) whose goal is to teach a robot to push an object to a predetermined target location. As in the experiments here, EI-FN outperforms other methods significantly, including EI-CF.

## 5.3 Calibration of an Epidemic Model

Here we consider calibration of compartmental stochastic models to data, a widely-used tool in epidemiology, medical modeling, and ecology (Sandberg, 1978). Function networks are well-suited to exploit the structure of such models. We focus on calibration of a specific epidemic model for influenza, building toward a COVID-19 mitigation benchmark in the next section. We first describe the epidemic model, then the calibration problem and, finally, formulation as a function network.

**SIS Epidemiological Model:** We calibrate to data a widely used epidemiological model, the SIS model (see, e.g., Garnett 2002), that models diseases like influenza capable of reinfecting individuals multiple times. In this model, individuals either do not have the disease and are "susceptible" (S) or

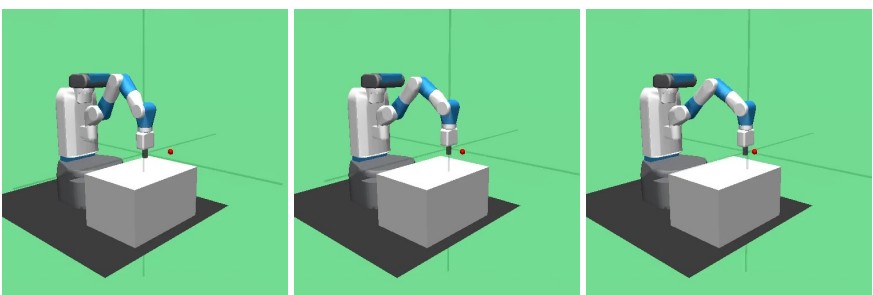

Figure 4: A sequence of screenshots showing three consecutive movements performed by the robotic arm described in §5.2.

have the disease and are "infectious" (I). We consider a SIS model of two interacting populations, where infections occur at population-dependent rates.

The model is dynamic, with time indexed by $t = 0, 1 \ldots, T$. At the beginning of each time period $t$, the fraction of population $i \in \{0, 1\}$ that is infectious is $I_{i,t}$. We assume each population is of equal size, $N$. During this time period, each person in group $i$ comes into close physical contact with $\beta_{i,j,t}$ people from group $j$. When this contact is between an infectious person and a susceptible one, it infects the susceptible person. A fraction $I_{j,t}$ of the people from group $j$ involved in such interactions are infectious and a fraction $(1 - I_{i,t})$ from group $i$ are susceptible. A number of new infections in group $i$ result, $N(1 - I_{i,t})\beta_{i,j,t}I_{j,t}$. As a fraction of group $i$'s population, this is $(1 - I_{i,t})\beta_{i,j,t}I_{j,t}$. Summing across $j$, we have $(1 - I_{i,t})\sum_j \beta_{i,j}(t)I_{j,t}$ new group $i$ infections. At the same time, infections resolve at a rate of $\gamma$ per period. This results in a decrease in the infectious population in group $i$ of $\gamma I_{i,t}$. Putting this together, the number of infectious individuals in group $i$ at the start of the next time period is $I_{i,t+1} = I_{i,t}(1 - \gamma) + (1 - I_{i,t})\sum_j \beta_{i,j,t}I_{j,t}$.

**Calibration:** The SIS model has parameters $I_{i,0}$, $\gamma$ and $\beta_{i,j,t}$, where $0 \le t < T$ and $i, j \in \{1, 2\}$. We calibrate the parameters $\vec{\beta} = (\beta_{i,j,t} : i, j, t)$ while fixing $\gamma = 0.5$ and $I_{i,0} = .01$ (for both $i$). We simulate a trajectory of infections from $t = 0$ up to $T = 3$, using a held-out value for $\vec{\beta}$. We let $I_{i,t}^{\text{obs}}$ denote the fraction of group $i$ observed to be infected at time $t$ in this trajectory. We then search for the vector $\vec{\beta}$ that, when passed to the SIS model, minimizes the mean squared error between this trajectory and the SIS model predictions. Letting $I_{i,t}(\vec{\beta})$ indicate this predicted value, the goal is to minimize the mean-squared error (MSE), $\text{MSE}(\vec{\beta}) := \sum_{t=1}^T (I_{i,t}^{\text{obs}} - I_{i,t}(\vec{\beta}))^2$. We do not include $t = 0$ since $I_{i,0}(\vec{\beta})$ is the same for all $\vec{\beta}$.

**Formulation as a Function Network:** We encode this as a function network using $2T + 1$ nodes, as illustrated in Figure 5. For each time period $t$ and each group $i$, a node takes input $I_t := (I_{j,t} : j = 0, 1)$ and $\beta_t := (\beta_{j,j',t} : j, j' \in \{0, 1\})$ and produces output $I_{i,t+1}$. (For $t = 0$, the input $I_0$ is not needed since this is the same for all $\vec{\beta}$.) Then, one additional node takes the output of the other nodes $(I_{i,t} : i = 0, 1, t = 1, \ldots, T)$ as its input and produces the sum of squared errors $\sum_{t=1}^T (I_{i,t}^{\text{obs}} - I_{i,t}(\vec{\beta}))^2$ as output. We treat this final node as known (its GP prior has a kernel of 0).

**EI-CF benchmark:** The fact that the final node in this problem (denoted "MSE" in Figure 5) has known structurre permits comparing against the EI-CF method for BO of composite functions (Astudillo and Frazier, 2019) as a benchmark. EI-CF is substantially less general than our method (EI-FN): it is restricted to settings with one time-consuming black-box multi-output node that provides input to one fast-to-evaluate node with known structure. To apply EI-CF to this problem, the single black-box multi-output node takes $\vec{\beta}$ as input and produces the vector $(I_{i,t}(\vec{\beta}) : i, t)$ as output. This output is then supplied to the "MSE" node. This approach ignores the fact that $I_{i,t}$ does not depend on $\beta_{t'}$, $t' > t$, and depends only indirectly on $\beta_{t'}$, $t < t$ through $I_{j,t-1}$, $j = 1, 2$.

## 5.4 Discussion

Across the wide range of problems considered, EI-FN significantly improves query efficiency over standard BO methods that ignore the function network structure of evaluations. The benefits range

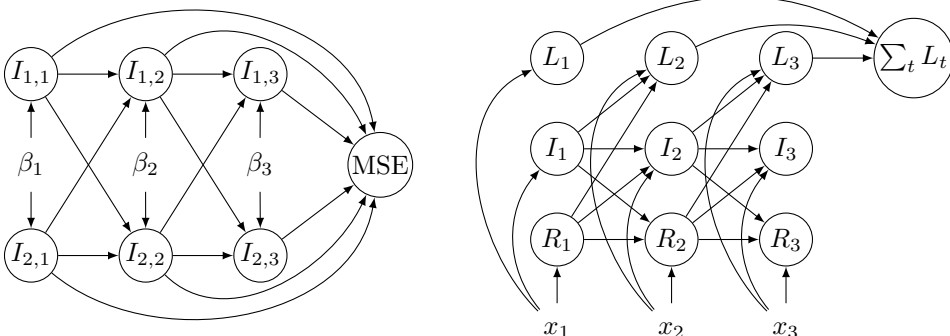

Figure 5: Function network for the (left) epidemic calibration problem in §5.3 and (right) the COVID-19 pooled testing optimization problem described in the supplement.

from a 5% improvement in the value of the best point found on the Drop-Wave and manufacturing problems to several orders of magnitude in the Rosenbrock and epidemic model calibration problems.

The largest benefit arise when the control input is high-dimensional but the input to individual nodes is low-dimensional. On the epidemic model calibration problem, we see EI-CF (Astudillo and Frazier, 2019) outperforming EI and KG by several orders of magnitude, and EI-FN outperforming EI-CF by several *additional* orders of magnitude. As noted above, EI-CF can be seen as a special case of EI-FN using a less informative function network that hides observations from some nodes. This is consistent with observations of function network structure allowing substantial improvement in query efficiency, and observing more of the internal function network structure providing more value.

## 6    Conclusion

We introduced a novel BO approach for objective functions defined by a series of expensive-to-evaluate functions, arranged in a network so that each function takes as input the output of its parent nodes. These objective functions arise in a wide range of application domains. However, existing methods cannot leverage this structure. Our approach models the outputs of the functions in this network instead of only the objective function, as is standard in BO. Our experiments show that, by doing so, this approach can dramatically outperform standard BO methods.

Though we see substantial benefits from our approach, there are limitations. First, it requires more computation than standard BO methods, as explored in the supplement. (When the objective is time-consuming, the improved query efficiency more than makes up for the additional computation required.) Second, while we have demonstrated our method in problems with up to 9 nodes, and computational speed would support more, our method does not (yet) scale to hundreds of nodes. Third, while we show consistency, it would be instructive to complement our empirical results showing fast convergence with a theoretical understanding of convergence rates. Existing approaches to prove convergence rates for the classical expected improvement heavily rely on properties of its analytical expression (Bull, 2011; Ryzhov, 2016), and thus are not directly generalizable to our setting. This is, however, an exciting direction for future work.

As with any powerful new method for optimizing time-consuming-to-compute black-box functions, ours can accelerate many applications. While this includes innovations that generally benefit society, such as improvements to public health and vaccine manufacturing, it also includes the design of weapons and other engineering systems that could harm individuals. Thus, it is important that society enact guardrails that ensure proper use of our methodology.

## Acknowledgments

The authors were partially supported by AFOSR FA9550-19-1-0283 and FA9550-20-1-0351.

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
