# Bayesian Optimization of Function Networks: Supplementary Material

**Raul Astudillo**
Cornell University
ra598@cornell.edu

**Peter I. Frazier**
Cornell University
pf98@cornell.edu

## A   Proof of Proposition 1

In this section, we provide a formal statement and proof of Proposition 1. We begin by proving the following auxiliary result.

**Lemma A.1.** *Suppose that $f : \mathbb{R}^{B_1 \times \ldots \times B_J} \to \mathbb{R}$ and $h_j : \mathbb{R}^A \to \mathbb{R}^{B_j}$, $j = 1, \ldots, J$, are all Lipschitz continuous with Lipschitz constants $L_f$ and $L_{h_j}$, $j = 1, \ldots, J$, respectively. Then, the function $g : \mathbb{R}^A \to \mathbb{R}$ defined by $g(x) = f(h_1(x), \ldots, h_J(x))$ is Lipschitz with Lipschitz constant $L_g := L_f \sum_{j=1,\ldots,J} L_{h_j}$.*

*Proof.* We have

$$
\begin{aligned}
|g(x) - g(x')| &= |f(h_1(x), \ldots, h_J(x)) - f(h_1(x'), \ldots, h_J(x'))| \\
&\leq L_f \|(h_1(x), \ldots, h_J(x)) - (h_1(x'), \ldots, h_J(x'))\|_2 \\
&\leq L_f \sum_{j=1}^{J} \|h_j(x) - h_j(x')\|_2 \\
&\leq L_f \sum_{j=1}^{J} L_{h_j} \|x - x'\|_2 \\
&= L_g \|x - x'\|_2.
\end{aligned}
$$

$\square$

We are now in position to show Proposition 1, which can be seen as a simple generalization of Theorem 1 in Balandat et al. (2020).

**Proposition A.1** (Proposition 1). *Suppose that $\mathbb{X}$ is compact, and that the functions $\mu_{n,k}, \sigma_{n,k} : \mathbb{R}^{|I(k)|} \times \mathbb{R}^{|J(k)|} \to \mathbb{R}$, $k = 1, \ldots, K$, are Lipschitz continuous. Let*

$$
\widehat{x}_*^{(M)} \in \underset{x \in \mathbb{X}}{\operatorname{argmax}} \, \widehat{\text{EI-FN}}_n \left( x; Z^{(1:M)} \right), \quad X_* = \underset{x \in \mathbb{X}}{\operatorname{argmax}} \, \text{EI-FN}_n(x),
$$

*where $\{Z_m\}_{m=1}^{\infty}$ are independent standard normal random variables. Then, for every $\epsilon > 0$, there exist $A, \alpha > 0$ such that $\mathbb{P}\left( \text{dist}\left( \widehat{x}_*^{(M)}, X_* \right) > \epsilon \right) \leq Ae^{-\alpha M}$ for all $M$.*

*Proof.* Let $L_{\mu_{n,k}}$ and $L_{\sigma_{n,k}}$ be the Lipschitz constants of $\mu_{n,k}$ and $\sigma_{n,k}$, respectively, and consider the functions $\widehat{f}_{n,k} : \mathbb{R}^{|I(k)|} \times \mathbb{R}^{|J(k)|} \times \mathbb{R} \to \mathbb{R}$, $k = 1, \ldots, K$, given by

$$
\widehat{f}_{n,k}(x_{I(k)}, y_{J(k)}, z_k) = \mu_{n,k}(x_{I(k)}, y_{J(k)}) + \sigma_{n,k}(x_{I(k)}, y_{J(k)}) z_k.
$$

35th Conference on Neural Information Processing Systems (NeurIPS 2021).

We note that, for any fixed $z_k$, the function $(x_{I(k)}, y_{J(k)}) \mapsto \widehat{f}_{n,k}(x_{I(k)}, y_{J(k)}, z_k)$ is $\left(L_{\mu_{n,k}} + L_{\sigma_{n,k}}|z_k|\right)$-Lipschitz.

Now consider the functions $\widehat{h}_{n,1}, \ldots, \widehat{h}_{n,k} : \mathbb{X} \times \mathbb{R}^K \to \mathbb{R}$ defined recursively by

$$\widehat{h}_{n,k}(x, z) = \widehat{f}_{n,k}\left(x_{I(k)}, \widehat{h}_{n,J(k)}(x, z), z_k\right), \quad k = 1, \ldots, K.$$

Applying Lemma 1 repeatedly, we find that, for any fixed $z \in \mathbb{R}^K$, the functions $x \mapsto \widehat{h}_{n,k}(x, z)$, $k = 1, \ldots, K$, are Lipschitz continuous with Lipschitz constants $L_{\widehat{h}_{n,k}}(z)$, $k = 1, \ldots, K$, defined recursively by

$$L_{\widehat{h}_{n,k}}(z) = \left(L_{\mu_{n,k}} + L_{\sigma_{n,k}}|z_k|\right)\left(1 + \sum_{j \in J(k)} L_{\widehat{h}_{n,j}}(z)\right), \quad k = 1, \ldots, K.$$

Let $\widehat{g}_n = \widehat{h}_{n,k}$ and $L_{\widehat{g}_n} = L_{\widehat{h}_{n,k}}$. Then, for any fixed $z$, the function $x \mapsto \widehat{g}_n(x, z)$ is $L_{\widehat{g}_n}(z)$-Lipschitz. Moreover, by definition, $\widehat{g}_n$ satisfies

$$\text{EI-FN}_n(x) = \mathbb{E}_n\left[\{\widehat{g}_n(x, Z) - g_n^*\}^+\right],$$

where $Z$ is a $K$-dimensional standard normal random vector and also

$$\widehat{\text{EI-FN}}_n\left(x; Z^{(1:M)}\right) = \frac{1}{M}\sum_{m=1}^{M}\left\{\widehat{g}_n\left(x, Z^{(m)}\right) - g_n^*\right\}^+.$$

Now observe that $L_{\widehat{g}_n}(z)$ is a polynomial in the variables $|z_1|, \ldots, |z_k|$ with degree at most 1 for each variable. Since the folded normal distribution has finite moment generating function everywhere Therefore, if $Z$ is $K$-dimensional standard normal random vector, then $L_{\widehat{g}_n}(Z)$ has a finite moment generating function in a neighborhood of 0.

An similar argument can be used to show that, for every $x$, $\widehat{g}_n(x, Z)$ has a finite moment generating function in a neighborhood of zero. The desired result is now a direct consequence of Proposition 2 in the supplement of Balandat et al. (2020), which is in turn a consequence of Theorem 2.3 in Homem-de Mello (2008). □

# B Proof of Theorem 1

In this section, we prove Theorem 1. Throughout this section, we let $(x_n : n)$ denote the sequence of points at which the function network is evaluated. We begin by introducing the following definition, which is analogous to Definition 2.1 in Bect et al. (2019).

**Definition B.1.** *Let $\mathcal{F}_n$ be the sigma algebra generated by the function network observations up to time $n$. The sequence $(x_n : n)$ is said to be a (non-randomized) sequential design if $x_n$ is $\mathcal{F}_{n-1}$-measurable for all $n$.*

Throughout this section, we assume that $(x_n : n)$ is a sequential design. We note, in particular, that if $x_n \in \max_{x \in \mathbb{X}} \text{EI-FN}_{n-1}(x)$ for all $n$, then $(x_n : n)$ is a sequential design.

Our proof relies on the following assumptions.

**Assumption B.1.** $\mathbb{X}$ *is compact.*

**Assumption B.2.** *The prior mean and covariance functions of $f_1, \ldots, f_K$, are such that $f_1, \ldots, f_K$ are continuous almost surely.*

**Assumption B.3.** *The prior covariance functions of $f_1, \ldots, f_K$ are bounded.*

**Assumption B.4.** *With probability 1 under the prior, given $f_1, \ldots, f_K$ and a sequential design $(x_n : n)$, there exists a function $\beta(z)$ such that $|\widehat{g}_{n-1}(x_n, z)| \leq \beta(z)$ for all $n$ and $z$ and $\int_{-\infty}^{\infty} \varphi(z)\beta(z) < \infty$, where $\varphi$ is the standard normal pdf.*

Assumptions B.1, B.2, and B.3 are standard. Assumption B.4 is bespoke to our arguments, but holds, for example, when the posterior mean of $f_K$ is uniformly bounded. (This bound can depend on $f_1, \ldots, f_K$.) This occurs, for example, when each $f_k$ is in the reproducing kernel Hilbert space (RKHS) corresponding to the prior covariance function. In particular, if $f_K$ is in this RKHS and the prior kernel is bounded, then $f_K$ is bounded and there is a uniform bound (depending on the RKHS norm of $f_K$ and the prior covariance) over the deviation between $f_K$ and the sequence of posterior means resulting from our observations. The sum of these two bounds and a term that is linear in $z$ arising from the posterior standard deviation term in the definition of $\widehat{g}_{n-1}(x_n, z)$ provides $\beta(z)$. We also believe that Assumption B.4 holds more broadly.

Note that our proof does not rely on the no-empty-ball assumption (NEB) of Vazquez and Bect (2010). Thus, our proof also extends the proof of Astudillo and Frazier (2019) to a broader class of prior distributions.

As in the main paper, we refer to the "time-$n$" posterior, which is the conditional distribution of $f_1, \ldots, f_K$ given $(x_m : m \le n)$ and $(h_k(x_m) : k \le K, m \le n)$. $\mathbb{E}_n$ denotes expectation with respect to this conditional distribution, and $\mathbb{P}_n$ denotes the probability operator.

Recall the sampling procedure from Section 4.3 of the main paper. This defined a function $\widehat{g}(x, Z)$ that depended on the current posterior distribution, a point $x$ in the feasible domain, and a vector $Z$. When $Z$ was generated as a standard normal random variable, the distribution of $\widehat{g}(x, Z)$ was the same as that of $g(x)$ under the current posterior. To support working with this over a sequence of posterior distributions, we use $\widehat{g}_n(x, Z)$ to indicate this function calculated using the posterior at time $n$. We similarly use the notation $\widehat{h}_k(x, Z)$ to represent the function $\widehat{h}_{n,k}(x, Z)$ from the main paper computed with respect to the posterior at time $n$.

**Lemma B.1.** $g_\infty^* := \lim_n g_n^*$ *exists and is finite almost surely. Moreover, any limit point $x_\infty$ of the sequence $(x_n : n)$ satisfies $g(x_\infty) \le g_\infty^*$.*

*Proof.* The sequence $(g_n^* : n)$ is non-decreasing and bounded above by the random variable $g^* := \max_{x' \in \mathbb{X}} g(x')$. The random variable $g^*$ is almost surely finite since $g$ is almost surely continuous (it is the composition of a collection of almost surely continuous functions $h_k$) and $\mathbb{X}$ is compact. Thus $g_\infty^* := \lim_n g_n^*$ exists and is finite almost surely.

Let $x_\infty$ be the limit of a convergent subsequence $(x_{n_m} : m)$ of $(x_n : n)$. Since $g$ is almost surely continuous,

$$g(x_\infty) = g(\lim_m x_{n_m}) = \lim_m g(x_{n_m}) \le \lim_m g_{n_m+1}^* \le \lim_m g_\infty^* = g_\infty^*.$$

$\square$

**Lemma B.2.** *Consider the almost sure event that $f_k$ is continuous for all $k = 1, \ldots, K$. On this event, the function $h_k$ is continuous for all $k = 1, \ldots, K$.*

*Proof.* We show this via induction. The base case, for $k = 1$, follows since $f_1$ is continuous on the event considered, $x_{I(k)}$ is a continuous function of $x$, and so $h_1(x) = f_1(x_{I(k)})$ is the composition of two continuous functions and so is a continuous function of $x$.

We now show the induction step. Fix $k > 1$. Suppose $h_{k'}(x)$ is continuous for all $k' < k$. applying the induction hypothesis for all $k' \in J(k) \subseteq \{1, \ldots, k-1\}$ implies that $h_{J(k)}(x)$ is continuous. Also $x_{I(k)}$ is a continuous function of $x$. Thus, $x \mapsto (x_{I(k)}, h_{J(k)}(x))$ is continuous. This and the fact that $f_k$ is continuous on the event considered implies that $h_k(x) = f_k(x_{I(k)}, h_{J(k)}(x))$ is a composition of continuous functions and so is continuous. $\square$

**Lemma B.3.** *For each $k = 1, \ldots, K$, the functions $\mu_{n,k}$ and $\sigma_{n,k}$ converge pointwise to some continuous functions $\mu_{\infty,k}$ and $\sigma_{\infty,k}$ almost surely; moreover, this convergence is uniform over compact subsets of $\mathbb{R}^{|I(k)|} \times \mathbb{R}^{|J(k)|}$.*

*Proof.* Fix $k$ and consider the almost sure event that $f_1, \ldots, f_{k-1}$ are continuous. Condition on continuous realizations of $f_1, \ldots, f_{k-1}$, thus fixing $h_{J(k)}$ as well.

Let $A$ be an arbitrary compact subset of $\mathbb{R}^{|I(k)|} \times \mathbb{R}^{|J(k)|}$ and define $B = \{(x_{I(k)}, h_{J(k)}(x)) : x \in \mathbb{X}\}$. $B$ is compact by Lemma B.2 since $x \mapsto (x_{I(k)}, h_{J(k)}(x))$ is continuous, $\mathbb{X}$ is compact, and the image

of a compact set through a continuous function is compact. The observations of $f_k$ occur at input points $\{(x_{n,I(k)}, h_{J(k)}(x_n)) : n\} \subset B$.

Let $C = A \cup B$ and note that $C$ is compact. Then, by Proposition 2.9 in Bect et al. (2019), $\mu_{n,k}$ and $\sigma_{n,k}$ converge uniformly over $C$ and thus also over $A$. $\qquad\square$

**Lemma B.4.** *Let $(x_{n_m} : m)$ be a convergent subsequence of $(x_n : n)$ with limit $x_\infty$. Then, $\lim_{m\to\infty} \widehat{g}_{n_m-1}(x_{n_m}, z) = g(x_\infty)$ for each $z \in \mathbb{R}^K$ almost surely.*

*Proof.* All the convergence claims made in this proof are almost surely. First note that Lemma B.3 implies that, for each $k = 1, \ldots, K$, the function $\widehat{f}_{n,k}$ defined in the proof of Proposition 1 converges to the function $\widehat{f}_{\infty,k}$ defined by

$$\widehat{f}_{\infty,k}(x_{I(k)}, y_{J(k)}, z_k) = \mu_{\infty,k}(x_{I(k)}, y_{J(k)}) + \sigma_{\infty,k}(x_{I(k)}, y_{J(k)})z_k.$$

uniformly in $(x_{I(k)}, y_{J(k)})$ (but not necessarily $z_k$). This in turn implies that, for each $k = 1, \ldots, K$, $\widehat{h}_{n,k}$ converges to the function $\widehat{h}_{\infty,k}$ defined recursively by

$$\widehat{h}_{\infty,k}(x, z) = \widehat{f}_{\infty,k}\left(x_{I(k)}, \widehat{h}_{\infty,J(k)}(x, z), z_k\right)$$

uniformly in $x \in \mathbb{X}$ for each $z \in \mathbb{R}^K$.

We show the following two claims by induction on $k$:

1. $\sigma_{n_m-1,k}(x_{n_m,I(k)}, \widehat{h}_{n_m-1,J(k)}(x_{n_m}, z)) \to 0$.

2. $\widehat{h}_{n_m-1,k}(x_{n_m}, z) \to h(x_\infty)$.

We first show the induction step, where we assume the induction hypothesis is true for all $k' < k$ and show it for $k$.

Each element of $J(k)$ is strictly less than $k$ and so the induction hypothesis implies that $\widehat{h}_{n_m-1,J(k)}(x_{n_m}, z) \to h_{J(k)}(x_\infty)$. Moreover, since $x_{n_m} \to x_\infty$, and $\sigma_{n,k}$ converges uniformly to $\sigma_{\infty,k}$, it follows that $\sigma_{n_m-1,k}(x_{n_m,I(k)}, \widehat{h}_{n_m-1,J(k)}(x_{n_m}, z))$ converges to $\sigma_{\infty,k}(x_{\infty,I(k)}, h_{J(k)}(x_\infty))$. Similarly, $\widehat{h}_{n_m-1,k}(x_{n_m}, z)$ converges to

$$\begin{aligned}
\widehat{h}_{\infty,k}(x_\infty, z) &= \widehat{f}_{\infty,k}\left(x_{\infty,I(k)}, \widehat{h}_{\infty,J(k)}(x_\infty, z), z_k\right) \\
&= \widehat{f}_{\infty,k}\left(x_{\infty,I(k)}, h_{J(k)}(x_\infty), z_k\right) \\
&= \mu_{\infty,k}(x_{\infty,I(k)}, h_{J(k)}(x_\infty)) + \sigma_{\infty,k}(x_{\infty,I(k)}, h_{J(k)}(x_\infty))z_k,
\end{aligned}$$

where the second equation is obtained by noting that, since $\widehat{h}_{n_m-1,J(k)}(x_{n_m}, z) \to h_{J(k)}(x_\infty)$ (by the induction hypothesis) and $\widehat{h}_{n_m-1,J(k)}(x_{n_m}, z) \to \widehat{h}_{\infty,J(k)}(x, z)$, it must be the case that $\widehat{h}_{\infty,J(k)}(x_\infty, z) = h_{J(k)}(x_\infty)$.

Now observe that $\sigma_{n_m+1,k}(x_{n_m,I(k)}, h_{J(k)}(x_{n_m}))$ converges to $\sigma_{\infty,k}(x_{\infty,I(k)}, h_{J(k)}(x_\infty))$, but $\sigma_{n_m+1,k}(x_{n_m,I(k)}, h_{J(k)}(x_{n_m})) = 0$ for all $m$, and thus $\sigma_{\infty,k}(x_{\infty,I(k)}, h_{J(k)}(x_\infty)) = 0$. This proves the first part of the induction step. Similarly, note that $\mu_{n_m+1,k}(x_{n_m,I(k)}, h_{J(k)}(x_{n_m}))$ converges to $\mu_{\infty,k}(x_{\infty,I(k)}, h_{J(k)}(x_\infty))$, but

$$\begin{aligned}
\mu_{n_m+1,k}(x_{n_m,I(k)}, h_{J(k)}(x_{n_m})) &= f_k(x_{n_m,I(k)}, h_{J(k)}(x_{n_m})) \\
&= h_k(x_{n_m}) \to h_k(x_\infty).
\end{aligned}$$

This proves the second part of the induction step.

The proof of the base case ($k = 0$) is analogous except that $J(k) = \emptyset$ eliminates terms that depend on $k' < k$.

$\qquad\square$

**Lemma B.5.** $\liminf_n \text{EI-FN}_{n-1}(x_n) = 0$ *almost surely.*

*Proof.* Since $(x_n : n)$ is contained in a compact set, it has a convergent subsequence, $(x_{n_m} : m)$. Then, letting $\varphi(z)$ be the standard normal probability density function,

$$\lim_{m \to \infty} \text{EI-FN}_{n_m - 1}(x_{n_m}) = \lim_{m \to \infty} \int_{-\infty}^{\infty} \left\{ \widehat{g}_{n_m - 1}(x_{n_m}, z) - g^*_{n_m - 1} \right\}^+ \varphi(z) \, dz$$

$$= \int_{-\infty}^{\infty} \lim_{m \to \infty} \left\{ \widehat{g}_{n_m - 1}(x_{n_m}, z) - g^*_{n_m - 1} \right\}^+ \varphi(z) \, dz$$

by the dominated convergence theorem and Assumption B.4.

By Lemmas B.1 and B.4,

$$\lim_{m \to \infty} \left\{ \widehat{g}_{n_m - 1}(x_{n_m}, z) - g^*_{n_m - 1} \right\}^+ = \left\{ g(x_\infty) - g^*_\infty \right\}^+ = 0$$

for each $z$. Thus, $\lim_{m \to \infty} \text{EI-FN}_{n_m - 1}(x_{n_m}) = 0$, implying that $\liminf_n \text{EI-FN}_{n-1}(x_n) \leq 0$. This and the fact that $\text{EI-FN}_{n-1}(x) \geq 0$ for all $x$ implies that $\liminf_n \text{EI-FN}_{n-1}(x_n) = 0$. $\qquad\square$

The following lemma considers a sequence of random variables $I_n$ that we will later take to be the random improvements generated within our statistical model under the posterior after $n$ measurements. The quantity $\mathbb{E}[I_n^+]$ will then be the EI-FN under this posterior.

**Lemma B.6.** *Consider a sequence of scalar random variables $I_n$. If $\liminf_n \mathbb{P}(I_n \geq \epsilon) > 0$ for any given $\epsilon > 0$, then $\liminf_n \mathbb{E}[I_n^+] > 0$.*

*Proof.* We have $\mathbb{E}[I_n^+] \geq \epsilon \mathbb{P}(I_n \geq \epsilon)$. Thus, $\liminf_n \mathbb{E}[I_n^+] \geq \epsilon \liminf_n \mathbb{P}(I_n \geq \epsilon) > 0$. $\qquad\square$

We are now ready to prove Theorem 1.

*Proof of Theorem 1.* Pick any point $x \in \mathbb{X}$. Since we choose to evaluate at the point with largest $\text{EI-FN}_n(x)$, $\text{EI-FN}_n(x) \leq \text{EI-FN}_n(x_{n+1})$ for each $n$.

Lemma B.5 then implies that there is a subsequence $(n_m : m)$ on which $\lim_m \text{EI-FN}_{n_m}(x_{n_m+1}) = 0$. This and the fact that $\text{EI-FN}_n(x) \geq 0$ imply that $\lim_n \text{EI-FN}_{n_m}(x) = 0$. Thus, $\liminf_n \text{EI-FN}_n(x) = 0$.

Recall that $\text{EI-FN}_n(x) = \mathbb{E}_n[\{g(x) - g_n^*\}^+]$. Consider the conditional distribution of $g(x) - g_n^*$ under the time-$n$ posterior. This is the same as the conditional distribution of $\widehat{g}_n(x; Z) - g_n^*$ where only $Z$ is random and the other quantities are completely determined by the observations of the function network at $x_1, \ldots, x_n$. By taking $I_n$ to be a random variable with the same distribution for each $n$, then on any sequence of observations, Lemma B.5 and the contrapositive of Lemma B.6 imply that $\liminf_n \mathbb{P}_n(g(x) - g_n^* \geq \epsilon) = 0$ for each $\epsilon > 0$.

Since the random variable $g_\infty^*$ defined in Lemma B.1 bounds each $g_n^*$ above, $\mathbb{P}_n(g(x) - g_n^* \geq \epsilon) \geq \mathbb{P}_n(g(x) - g_\infty^* \geq \epsilon)$ and we have $\liminf_n \mathbb{P}_n(g(x) - g_\infty^* \geq \epsilon) = 0$ for each $\epsilon > 0$.

For any event $W$, $(\mathbb{P}_n(W) : n)$ is a uniformly integrable martingale, and thus converges almost surely to a limiting random variable $\mathbb{P}_\infty(W)$, where $\mathbb{P}_\infty$ is defined as the conditional expectation with respect to the event $(x_n, h_k(x_n) : n < \infty, k \leq K)$ (by Theorem 5.13 of Çınlar (2011)). Taking $W$ to be the event that $g(x) - g_\infty^* \geq \epsilon$, we have that $\mathbb{P}_n(g(x) - g_\infty^* \geq \epsilon)$ has a limit, $\mathbb{P}_\infty(g(x) - g_\infty^* \geq \epsilon)$. Moreover, this limit must be the same as the $\liminf$, which we showed above was 0. Thus,

$$\mathbb{P}_\infty(g(x) - g_\infty^* \geq \epsilon) = 0.$$

Since this is true for each $\epsilon > 0$, taking the limit as $\epsilon \to 0$ and using the monotone convergence theorem shows

$$\mathbb{P}_\infty(g(x) > g_\infty^*) = 0.$$

Taking the expectation under the prior and applying the law of conditional expectation, we have that

$$0 = \mathbb{E}\left[\mathbb{P}_\infty(g(x) > g_\infty^*)\right] = \mathbb{E}\left[\mathbb{E}_\infty(1\{g(x) > g_\infty^*\})\right] = \mathbb{E}\left[1\{g(x) > g_\infty^*\}\right] = \mathbb{P}(g(x) > g_\infty^*).$$

Thus, the value of $g(x)$ is almost surely less than or equal to the limiting value of the sequence of best points found.

Let $X$ be a countable set that is dense in $\mathbb{X}$. Such set exists because $\mathbb{X}$ is compact. Then, because the countable union of events with probability zero also has probability zero,

$$0 = \mathbb{P}(g(x) > g_\infty^* \text{ for some } x \in X) = \mathbb{P}\left(\sup_{x \in X} g(x) > g_\infty^*\right)$$

Moreover, because $g$ is almost surely continuous and $X$ is dense in $\mathbb{X}$, $\sup_{x \in \mathbb{X}} g(x) = \sup_{x \in X} g(x)$ almost surely. Hence, $\mathbb{P}\left(\sup_{x \in \mathbb{X}} g(x) > g_\infty^*\right) = 0$, which concludes the proof. $\qquad\square$

## C  Proof of Proposition 2

In this section we prove Proposition 2 by providing a function network and a set of initial conditions where EI-FN does not measure the optimization domain densely. While the example we provide is very simple, we think such behavior also arises in more complex networks.

**Proposition C.1** (Proposition 2). *There exists a function network in which EI-FN is consistent but whose measurements are not necessarily dense in $\mathbb{X}$.*

*Proof.* Let $\mathbb{X} = [0, 1]$ and consider a function network with two nodes $f_1 : \mathbb{X} \to \mathbb{R}$ and $f_2 : \mathbb{X} \times \mathbb{R} \to \mathbb{R}$ where $f_2$ is deterministic, given by $f_2(x, y) = \max\{1, y\} - x$, and the objective function is given by $g(x) = f_2(x, f_1(x))$.

Suppose that $f_1$ is drawn from a GP prior with a continuous mean function and a bounded positive definite covariance function whose sample paths are almost surely continuous. From this and the fact that $f_2$ is deterministic and continuous, it follows that Assumptions 2.1-2.3 in §2 are satisfied. Assumption 2.4 is also satisfied because $f_2$ is bounded over $\mathbb{X} \times \mathbb{R}$. Thus, Theorem 1 implies that EI-FN is consistent almost surely.

Let $\tau = \inf\{n \geq 1 : f_1(x_n) > 1\}$ be the first time that we measure a point whose value for $f_1$ is strictly greater than 1. (If we never measure such a point, then $\tau$ is infinity.)

If $\mathbb{P}(\tau < \infty) = 0$ under EI-FN, then this problem is one in which EI-FN does not measure densely. This is because there is a strictly positive probability that $\sup_{x \in [0,1]} f_1(x) > 1$. Moreover, the fact that $f_1$ is almost surely continuous implies that on this event there is an non-empty interval on which $f_1(x)$ is strictly above 1 over the entire interval. If EI-FN were to never measure in this interval then it would not have measured densely. Thus, going forward, we assume $\mathbb{P}(\tau < \infty) > 0$. In fact, using a similar argument, we may assume $\mathbb{P}(\tau < \infty, \ f_1(0) < 1, \ f_1(1) < 1) > 0$.

For any $x > x_\tau$, we have $g(x) \leq 1 - x < 1 - x_\tau = g(x_\tau)$ almost surely. Thus, any such $x$ is almost surely strictly suboptimal under the posterior and EI-FN$_n(x) = 0$ for all $n \geq \tau$.

Now let $n \geq \tau$ and consider any unmeasured point $x$ with $1 - x > g_n^*$; such a point exists on the event under consideration since $f_1(0) < 1$ implies $g_n^* \leq \max\{f_1(0), 1 - \min_{m \leq n} x_m\} < 1$ and we make take $x$ arbitrarily close to 0. Because the prior covariance function is positive definite, the posterior probability distribution over $f_1(x)$ has full support over the real line. Thus, in particular, there is a strictly positive posterior probability that $f_1(x) \geq 1$. On this event, $g(x) = 1 - x > g_n^*$ and so EI-FN$_n(x)$ is strictly positive.

It follows that EI-FN would not measure at a point in the interval $(x_\tau, 1]$, which concludes the proof.

$\qquad\square$

## D  Additional Details on the Numerical Experiments

### D.1  Hyperparameter Estimation, Number of q-MC Samples, Runtimes, and Licenses

All GPs in our experiments have a constant mean function and ARD Matérn covariance function with smoothness parameter equal to 5/2, which is a standard choice in practice. The length scales of these GPs are estimated via maximum a posteriori (MAP) estimation with Gamma priors.

We use $M = 128$ quasi-MC samples obtained via scrambled Sobol sequences (see Balandat et al. (2020) for details) for computing the SAA of EI-FN. We use the same number of samples for EI-CF

and 8 samples for KG. KG is maximized following the one-shot approach proposed introduced by Balandat et al. (2020). Under this approach, the dimension of the optimization problem that arises when optimizing KG grows linearly with the number of samples and thus one is restricted to a small number of samples. The average runtimes of the BO methods for each of the problems are summarized in Table 1. We emphasize that, while optimizing EI-FN is more expensive than optimizing EI, the additional computation required by our method is compensated by its excellent performance, and is thus justified for problems where each function network evaluation takes several minutes or more. We also note KG becomes very expensive to optimize for problems with relatively high input dimension, and can be even more expensive than EI-FN.

The BoTorch python package and the source code for the robot pushing problem are both publicly available under a MIT licence. Our code is also publicly available under a MIT license.

Table 1: Average runtimes (seconds) per evaluation of the BO methods compared. EI-CF is N/A in problems that lack the structure it requires for use: that the objective is a composition of an inner black-box function and a outer known non-linear function.

|  | KG | EI | EI-CF | EI-FN |
|---|---|---|---|---|
| Drop-Wave | 15.1 | 2.5 | N/A | 15.4 |
| Rosenbrock, $K = 4$ | 23.6 | 4.16 | N/A | 122.2 |
| Ackley | 72.1 | 18.3 | N/A | 89.2 |
| Alpine2, $K = 6$ | 84.1 | 22.5 | N/A | 215.6 |
| Manufacturing | 29.1 | 5.4 | N/A | 117.2 |
| COVID-19 | 43.1 | 6.2 | N/A | 229.4 |
| Robot | 935.2 | 29.6 | 110.1 | 182.3 |
| Calibration | 1225.2 | 43.5 | 207.2 | 293.7 |

## D.2 Details on Synthetic Test Problems

Here we describe in detail how each of the synthetic test functions is arranged as a function network.

### D.2.1 Alpine2

The Alpine2 test function (Jamil and Yang, 2013) is defined by

$$g(x) = -\prod_{k=1}^{K} \sqrt{x_k} \sin(x_k).$$

We adapt this function to our setting by letting

$$f_1(x_1) = -\sqrt{x_1} \sin(x_1);$$

$$f_k(x_k, y_{k-1}) = \sqrt{x_k} \sin(x_k) y_{k-1}, \ k = 2, \ldots, K;$$

$I(k) = \{k\}, \ k = 1, \ldots, K; \ J(1) = \emptyset;$ and $J(k) = \{k-1\}, \ k = 2, \ldots, K$. In our experiments, we set $\mathbb{X} = [0, 10]^K$, and consider $K = 2, 4,$ and $6$.

The network structure of this test function can be summarized as a series of nodes where the output of each node is governed by one decision variable of its own, and the output of the previous node.

### D.2.2 Ackley

The Ackley test function (Jamil and Yang, 2013) has been widely used as a benchmark function in the BO literature. It is defined by

$$g(x) = 20 \exp\left(-0.2\sqrt{\frac{1}{D}\sum_{d=1}^{D} x_d^2}\right) + \exp\left(\frac{1}{D}\sum_{d=1}^{D} \cos(2\pi x_d)\right) - 20 - e.$$

We adapt it to our setting by letting

$$f_1(x) = \frac{1}{D} \sum_{d=1}^{D} x_d^2;$$

$$f_2(x) = \frac{1}{D} \sum_{d=1}^{D} \cos(2\pi x_d);$$

$$f_3(y_1, y_2) = 20 \exp\left(-0.2\sqrt{y_1}\right) + \exp\left(y_2\right) - 20 - e;$$

$I(1) = I(2) = \{1, \ldots, D\}; I(3) = \emptyset; J(1) = J(2) = \emptyset;$ and $J(3) = \{1, 2\}$. In our experiment, we set $\mathbb{X} = [-2, 2]^D$, and $D = 6$.

### D.2.3   Rosenbrock

The Rosenbrock test function (Jamil and Yang, 2013) is also a widely used benchmark function in the BO literature. It is defined by

$$g(x) = -\sum_{d=1}^{D-1} 100(x_{d+1} - x_d^2)^2 + (1 - x_d)^2.$$

We adapt it to our setting by letting

$$f_1(x_1, x_2) = -100(x_2 - x_1^2)^2 - (1 - x_1)^2;$$

$$f_k(x_k, x_{k+1}, y_{k-1}) = -100(x_{k+1} - x_k^2)^2 - (1 - x_k)^2 + y_{k-1}, \ k = 2, \ldots, D - 1;$$

$I(k) = \{k, k+1\}, \ k = 1, \ldots, D - 1; J(1) = \emptyset;$ and $J(k) = \{k - 1\}, \ k = 2, \ldots, D - 1$. In our experiments, we set $\mathbb{X} - [-2, 2]^D$, and consider $D = 3, 5, \ and \ 7$.

### D.2.4   Drop-Wave

The Drop-Wave test function (Surjanovic and Bingham, 2013) is highly multi-modal and complex. It is defined by

$$g(x) = \frac{1 + \cos\left(12\sqrt{x_1^2 + x_2^2}\right)}{2 + 0.5\left(x_1^2 + x_2^2\right)}.$$

We adapt it to our setting by taking

$$f_1(x) = \sqrt{x_1^2 + x_2^2};$$

$$f_2(y_1) = \frac{1 + \cos\left(12y_1\right)}{2 + 0.5y_1^2};$$

$I(1) = \{1, 2\}; I(2) = \emptyset; J(1) = \emptyset;$ and $J(2) = \{1\}$. In our experiment, we set $\mathbb{X} = [-5.12, 5.12]^2$.

### D.3   Manufacturing Throughput Maximization

Here we describe the manufacturing throughput manufacturing test problem. This problem is similar in spirit to the biomanufacturing example in the introduction, but focusing on more traditional manufacturing in which workproduct is discrete rather than continuous. We have a manufacturing line with a series of stations that perform operations: e.g., steel is cut to size, then bent to shape, then holes are drilled, and finally the piece is painted. We consider "make-to-order' in which custom features of the part (e.g., color, size, orientation of the holes) require waiting until a customer order arrives to begin processing the part.

Orders for parts arrive randomly according to a homogeneous Poisson process. Orders move to the first station in the manufacturing line and enter a queue where they wait. The first order in the queue requires a processing time exponentially distributed with a service rate that decreases with the amount of resource devoted to that station (e.g., more workers, better machines). Parts not being processed wait in the queue until they arrive to the front of the queue. Once processed, a part moves to the second station where it similarly waits in a queue until it is at the front, then waits an exponential

amount of time that depends on a second service rate, which we can again control through staffing. This continues, with each part completing service moving to the next queue until it has completed service at all stations.

Our goal is to choose a collection of service rates, one for each station, to maximize the number of parts finished in a fixed amount of time. We constrain the sum of the service rates across the stations to represent a limit on total resources that can be allocated.

In our experiment, we consider a manufacturing line with 4 stations. The objective to maximize is the throughput of the network in steady state, $f(x)$, where $x_i$ is the service rate of station $i$, over the feasible domain $\mathbb{X} = \{x : 0 \leq x_i, \ i = 1, ..., 4, \ \text{and} \ \sum_{i=1}^4 x_i \leq 1\}$. Let $f_i(x_i, y_{i-1})$ be throughput of station $i$ in steady state given that the service rate of station is $i$ is $x_i$ and station $i-1$ has throughput $y_{i-1}$ in steady state. Then, our objective function can be written as a function network by taking $I(k) = \{k\}$, $k = 1, \ldots, K$; $J(1) = \emptyset$; and $J(k) = \{k-1\}$, $k = 2, 3, 4$. This network is illustrated in Figure 1.

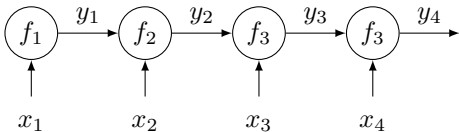

Figure 1: Manufacturing line as a function network.

## D.4   Optimization of Pooled Testing for COVID-19

Here we describe our COVID-19 testing benchmark problem, which considers reinforcement learning for large-scale testing to prevent the spread of COVID-19. It builds on the model for infection dynamics described in the epidemic model calibration problem in §5.3 of the main paper.

**Overview**   We design an asymptomatic screening protocol for controlling the spread of COVID-19 in a city. This approach regularly tests the entire population to identify people who are infected but do not have symptoms and may be unknowingly spreading virus. This approach has been employed successfully to control the spread of COVID-19 among students at several US universities (Denny et al., 2020) and also in Wuhan, China (Cao et al., 2020). To limit the resources needed for testing, we use *pooled testing*, which is described in detail in the supplement. This has a parameter (the *pool size*) that, when increased, reduces the testing resources used per person tested but also degrades test accuracy in a complex way that depends on the prevalence of the virus in the population.

We simulate the effect of asymptomatic screening using pooled testing on a single population, indexing time by $t = 1, 2, 3$. As in §5.3, we track the fraction of the population that is infectious and susceptible. To model the immunity that follows infection with COVID-19, an individual can be "recovered" (R), which means that they were previously infected, are no longer infectious and cannot be infected again. At the start of time period $t$, $I_t$ is the fraction of the population that is infectious, and $R_t$ is the fraction that is recovered. The rest is susceptible.

During each period $t$, the entire population is tested using a pool size of $x_t$. A black-box simulator determines the accuracy of these tests and the testing resources used (which depends on both $x_t$ and the prevalence $I_t$). Individuals testing positive are isolated[1] so that they cannot infect others during the period, and infectious individuals missed in testing infect others. Lower accuracy results in more individuals missed in testing. At the end of the period, all individuals in isolation are modeled as having recovered and leave isolation. This process results in a loss $L_t$ incorporating infections, testing resources used, and individuals isolated. Our goal is to choose the pool sizes $x_1$, $x_2$, $x_3$ to minimize the total loss $\sum_t L_t$.

This is encoded as a function network in Figure 3 in the main paper. As described above and in detail below, each time period performs a calculation that takes the pool size $x_t$ and a bivariate description $(I_t, R_t)$ of the population's current infection status. (The details of this computation is unknown to our function networks Bayesian optimization model.) It then produces as output a loss $L_t$ and the

---

[1]This is sometimes confused with quarantine: close contacts are quarantined, while positives are isolated

corresponding description $(I_{t+1}, R_{t+1})$ of the population's infection status at the start of the next period. The objective function is $\sum_t L_t$. The known form of the final node $\sum_t L_t$ is leveraged while the other nodes are treated as black boxes.

Given this overview of the problem, we now describe in detail how these black boxes are computed.

**Pooled Testing**   We first describe pooled testing in more detail. Pooled testing is a method for testing a large number of people for the presence of virus or some other pathogen (Cleary et al., 2021; Dorfman, 1943) in a way that reduces the amount of resource (specifically, chemical reagent and machine time) required per test performed compared with testing each person individually.

As in any COVID-19 test, we first collect a nasal or saliva sample from each individual being tested. Each sample is placed in a separate tube of fluid. Pooled testing relies on the ability to be able to test several people (a "pool") simultaneously, returning a signal that tells us whether (1) no one in the pool is positive; or (2) at least one person in the pool is positive. To accomplish this, a bit of fluid from each sample in the pool is taken and mixed together. Then a single chemical reaction (PCR, or *polymerase chain reaction*) is run to asses whether anyone in the pool is positive.

We specifically consider square array pooled testing (Westreich et al., 2008). This approach considers $x^2$ saliva samples as occupying a $x \times x$ grid. Then, it forms $2x$ pools: one pool from the samples in each row; and one from the samples in each column. Pools are tested, as described above. If a sample's row or column pool tests negative, then that sample is considered free of virus. All other samples (those whose row and column pool both test positive) are tested individually using a chemical reaction performed on additional fluid from that sample. This is illustrated in Figure 3 in the main paper.

The chemical reactions used to check for virus sometimes make errors: both false negatives, in which a pool or individual sample including material from an infected person tests negative; and false positives, in which a pool that does not contain virus nevertheless is deemed positive. Moreover, the probability of a false negative rises with the pool size (Cleary et al., 2021). This results in errors from the overall pooled testing procedure, where an individual who is virus-free is deemed positive (a false positive) or an individual who is infected with virus is deemed negative (a false negative).

In addition to depending on the pool size, the probability of these two kinds of errors (false positives and false negatives) in the overall testing procedure depends on the prevalence, i.e., the fraction of the population infected. When prevalence is high, there are sometimes two positive individuals providing fluid to a pool. This increases the chance that the pool tests positive. Also, more poools contain positive individuals, increasing the number of negative people whose row and column pools both test positive. This increases the chance that an overall test of a virus-free person will come back positive.

The level of resource used is proportional to the number of chemical reactions performed. This also depends on the pool size and the prevalence If the prevalence is small, then the number of chemical reactions used is approximately $1/(2x)$ since the number of chemical reactions performed on individual samples is small. As prevalence rises, larger pool sizes require more followup testing (because the pools become likely to contain at least one positive individual) and smaller pool sizes become efficient.

**Infection Dynamics without Pooled Testing**   As described above, time is divided into discrete time points $t = 1, 2, 3$, each representing a distinct two-week period. At the start of each period, the population is described by two numbers: $I_t$, the fraction of the population that is infectious; and $R_t$, the fraction of the population that is recovered and cannot be infected again. These numbers are both in $[0, 1]$. The additional $S_t = 1 - I_t - R_t$ fraction of the population is susceptible, and can be infected. Such divisions of a population into these three different groups (susceptible, infectious, and recovered) is widely used in epidemiology (Frazier et al., 2020).

We first describe our assumed infection dynamics in the absence of asymptomatic screening. This is obtained by integrating continuous-time dynamics within a given two-week period. We denote time strictly within a two-week period by $t + u$, where $t$ is an integer and $u \in (0, 1)$. During this period, we assume that infectious individuals who were infectious at the start of the period remain so for the full two weeks. Each comes into physical contact with other people at a rate of $\beta$ people per unit

time. A fraction $S_t$ are susceptible and become infected[2]. This gives us the differential equation

$$\frac{d}{ds} I_{t+u} = \beta S_t I_{t+u},$$

which has solution:

$$I_{t+u} = I_t \exp(\beta S_t u).$$

At the end of the two week period, we then assume that all individuals who were infectious at the start of the period convalesce and become recovered. Putting this together, in the absence of asymptomatic screening, the resulting dynamics would be:

$$S_{t+1} = S_t - I_t \exp(\beta S_t) \tag{1}$$
$$I_{t+1} = I_t(\exp(\beta S_t) - 1) \tag{2}$$
$$R_{t+1} = R_t + I_t \tag{3}$$

Although the details of these dynamics are different from those in §5.3, it results in behavior that is qualitatively similar. In particular, if $\beta$ is small enough, then $I_t$ shrinks to 0, but if it is large enough then the fraction of the population infected grows to a high fraction over a small number of time periods.

In our implementation, we set $\beta = (14/3)\ln(2) \approx 3.23$, corresponding to an epidemic that doubles in size every 3 days in the absence of any interventions.

We now incorporate the effect of asymptomatic screening.

**Infection Dynamics with Pooled Testing**  Our simulation includes pooled testing as follows. Pooled testing using the pool size $x_t$ is used at the start of the period. As described above, its error rates (false positive and false negative) and its efficiency depend on both the pool size and the prevalence ($I_t$). We use a black-box computation using logic described above to calculate three quantities:

- $\alpha_{FP}(x_t, I_t)$, the fraction of virus-free individuals tested that test positive (i.e., the false positive rate for the overall pooled testing procedure);
- $\alpha_{TP}(x_t, I_t)$, the fraction of infected individuals tested that test positive (i.e., the true positive rate for the overall pooled testing procedure);
- $\alpha_C(x_t, I_t)$, the number of chemical reactions performed across the entire population.

Individuals that test positive are immediately removed from the population placed into isolation. This includes both infectious individuals (in particular, a fraction $(\alpha_{TP}(x_t, I_t))I_t$ of the overall population) as well as susceptible and recovered individuals who were incorrectly classified (fractions $\alpha_{FP}(x_t, I_t)S_t$ and $\alpha_{FP}(x_t, I_t)R_t$ of the overall population respectively). Thus, the number of people isolated in period $t$ is,

$$Q_t = \alpha_{TP}(x_t, I_t)I_t + \alpha_{FP}(x_t, I_t)(S_t + R_t).$$

This results in a term $c_Q Q_t$ that is added to our loss, representing the social costs of isolation.

Because some infectious individuals are in isolation, the number of new infections is smaller than in the setting described above without asymptomatic screening. This number is $I_t(1 - \alpha_{TP}(x_t, I_t))$. Following the infection dynamics described above, this results in an additional new $I_t(1 - \alpha_{TP}(x_t, I_t)) \exp(\beta S_t)$ infections drawn from the susceptible population. In addition, all individuals who were infectious as the start of period $t$ recover. Thus, our dynamics are:

$$I_{t+1} = I_t(1 - \alpha_{TP}(x_t, I_t)) \exp(\beta S_t)$$
$$S_{t+1} = S_t - I_{t+1}$$
$$R_{t+1} = R_t + I_t$$

One may wonder about two modeling details. First, susceptible people who are erroneously in isolation are nevertheless modeled as eligible for infection. Additionally, recovered people are

---

[2]Note that we use $S_t$ rather than $S_{t+u}$. This allows for analytical solution to the above equation and does not substantially harm accuracy: in the regimes of importance for solving the benchmark problem optimally, $S_t$ begins close to 1 and $I_t$ begins close to 0.

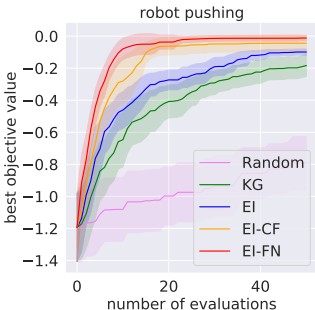

Figure 2: Results from the experiment described in §E.

modeled as being tested, although in practice one might choose to not test these individuals. These assumptions have little impact on outcomes because (1) false positive rates are small enough that the fraction of the susceptible population in isolation is a very small fraction of the overal susceptible population; (2) in the regimes where good solutions lie, few people are ever infected, making the recovered population also small. Making these assumptions simplifies the description and implementation.

The loss at time $t$ is the sum of the social cost of isolation described above, the cost of the testing supplies consumed $c_T \alpha_C(x_t, I_t)$, and the social cost associated with the new infections, $c_I I_{t+1}$.

$$L_t = c_T \alpha_C(x_t, I_t) + c_Q Q_t + c_I I_{t+1}.$$

## E    Additional Numerical Experiment: Active Learning for Robot Pushing

Here we describe one additional experiment. We consider a variation of the active learning for robot pushing problem introduced by Wang and Jegelka (2017) whose goal is to teach a robot to push an object to a predetermined target location. We modify the problem by allowing the robot to push the object several times instead of only once. We formalize this problem as follows. Let $x_{\text{init}}, x_{\text{target}} \in [-5, 5]^2$ denote the object's initial and target locations, respectively. At each time step, $t$, we choose the location of the robot's arm, $r_t \in [-5, 5]^2$, and the duration of the push, $d_t \in [1, 12]$. The robot then moves its arm from $r_t$ in the current direction of the object, $x_t$, over $d_t$ units of time. After this push, the location of the object becomes $x_{t+1}$ (if the robot fails to push the object, $x_{t+1} = x_t$). The goal is to choose $(r_t, d_t)$ for $t = 1, \ldots, T$ to minimize $\|x_{\text{target}} - x_T\|_2^2$. We set $x_{\text{init}} = (0, 0)$, $x_{\text{target}} = (2.9, 1.6)$, and $T = 3$. This can be interpreted as a function network by associating each time step with a pair of node functions $p_{t,1}$ and $p_{t,1}$ which take $(r_t, d_t, x_t)$ as input and produce $x_{t+1} = (p_{t,1}(r_t, d_t, x_t), p_{t,2}(r_t, d_t, x_t))$ as output.

The results of this experiment are shown in Figure 2. EI-FN improves over EI-CF and improves substantially over EI and Random.

## F    Posterior Mean and Covariance Functions

In this section, we write explicit formulas for the posterior mean and covariance functions of the GP distributions associated to the node functions, $f_1, \ldots, f_K$. To write this more simply, we define some additional notation. Given generic vectors $x \in \mathbb{R}^D$ and $y \in \mathbb{R}^K$, we define $z_k = (x_{I(k)}, y_{J(k)})$ as the elements of these vectors supplied as input to node $k$. Similarly, given a historical observation of the values of the node functions $y_\ell = (h_1(x_\ell), \ldots, h_K(x_\ell))$, $\ell = 1, \ldots, n$, we define $z_{\ell,k} = (x_{\ell,I(k)}, y_{\ell,J(k)})$. Using this notation, our posterior mean and covariance functions can be written as

$$\mu_{n,k}(z_k) = \mu_{0,k}(z_k) + \Sigma_{0,k}(z_k, z_{1:n,k}) \Sigma_{0,k}(z_{1:n,k}, z_{1:n,k})^{-1}(y_{1:n,k} - \mu_{0,k}(z_{1:n,k})),$$

and

$$\Sigma_{n,k}(z_k, z_k') = \Sigma_{0,k}(z_k, z_k') - \Sigma_{0,k}(z_k, z_{1:n,k}) \Sigma_{0,k}(z_{1:n,k}, z_{1:n,k})^{-1} \Sigma_{0,k}(z_{1:n,k}, z_k'),$$

respectively.