# OpenReview forum: "Bayesian Optimization of Function Networks"
_NeurIPS.cc/2021/Conference — NeurIPS 2021 Poster_

### Official Review · Reviewer_vzc8 · 2021-07-15

**Rating:** 5
**Confidence:** 5

**Summary:**

This paper focuses on Bayesian optimization of function networks, where the objective function has a known structure that is a composition of individual (node) functions that take the output of the previous function as input (along with potentially other parameters). Observations at each node function are obtained at each objective function evaluation. Such objective functions arise in a variety of applications including materials design and physical processes. Exploiting knowledge of the network structure can result in a lower effective input dimensionality of the model of each node function, depending on the network structure. By modeling each node function as distinct GP, the authors demonstrate improved sample efficiency using EI compared to optimizing the global objective function with a single GP or using compositional objectives.

**Limitations And Societal Impact:**

The discussion of the work's limitations is quite thorough, and it proposes interesting directions for future work. The authors have addressed potential negative societal impacts.

**Main Review:**

Strengths:
-	The idea that accounting for this structure is beneficial is simple and intuitive
- The empirical performance is quite compelling

Weaknesses:
-	There is not much technical novelty. Given the distinct GPs modeling the function network, the acquisition function and sampling procedure are not novel
-	The theoretical guarantee is pretty weak (random search is asymptotically optimal).
  - The discussion of not requiring dense coverage to prove the method is asymptotically consistent is interesting, but the utility of proposition 2 is not clear because although dense coverage is a consideration for proving consistency, it is not really a practical reality in sample-efficient optimization—typically BO would not have dense coverage.

Questions/comments:
-	There is no discussion of observation noise, which is a practical concern in many of the real world use cases mentioned in the paper. The approach of using GPs to model nodes in function network can naturally handle noisy observations, so only the acquisition function would need to be adjusted to account for noisy observations since the best objective value would be unknown. I expect that the empirical performance would remain the same (e.g. using Noisy EI from Letham et al. 2019), but the computation would be much more expensive. It would be good to discuss and demonstrated performance under noisy observations.
-	How does the number of MC samples affect performance, empirically? How does the network structure affect this?
-	It would be interesting to see a head-to-head comparison with deep GPs. How different are the runtimes (including inference times) and empirical performances?
- Since the core contribution is modeling each node in the function network with a distinct GP, it would be good to see more evaluation of the function network model's predictive performance compared to a alternative modeling choices (e.g. individual models with a compositional objective, vanilla global gp, deep gp)

Grammar:
-	L238 “out method” -> “our method”
-	L335 “structurre” -> “structure”


**Time Spent Reviewing:**

2

---

> ### Author Response · Authors · 2021-08-11
> **Author response to reviewer vzc8**
>
> We would like to thank you for your feedback and questions. We are glad that you found the empirical performance of our method “quite compelling”. Below, we respond to your questions and comments. We hope, in particular, that our discussion below of our theoretical results more clearly articulates the insight they provide, addressing the concerns listed in the review. We sincerely hope these answers allow you to consider increasing the rating of our paper.
>
> $\textbf{Q1:}$ The theoretical guarantee is pretty weak… The discussion of not requiring dense coverage to prove the method is asymptotically consistent is interesting, but the utility of proposition 2 is not clear because although dense coverage is a consideration for proving consistency, it is not really a practical reality in sample-efficient optimization—typically BO would not have dense coverage.
>
> $\textbf{A1:}$  We agree that asymptotic consistency is a rather modest theoretical guarantee. However, proving this without requiring dense evaluations is insightful for the following reason: The proof of Proposition 2 shows not only that measurements of our method are not necessarily dense, but also that our method is able to certify entire regions as suboptimal in finite time; no other method can do this. If a standard BO method or random search were applied in this problem instance, we would expect them to make at least a few subsequent evaluations in the region $(x_{\tau}, 1]$. In contrast, our method is able to certify this region as suboptimal and thus wastes no subsequent evaluations there. This ability provides insight into why our method can substantially outperform standard BO methods even if in practice these methods are not run for enough iterations for them to exhibit dense coverage.
>
> $\textbf{Q2:}$ There is not much technical novelty.
>
> $\textbf{A2:}$ The main novelty of our work lies in our function network model, which views a class of problems that is “highly relevant in practice (r59t)” in a way that enables dramatic improvements over the previous state of the art. As noted by other reviewers, we believe that our paper provides “useful learning to the general community (HdZa)”.
>
> $\textbf{Q3:}$ How does the number of MC samples affect performance, empirically? How does the network structure affect this?
>
> $\textbf{A3:}$ In preliminary experiments, we found that our method is fairly robust to the number of (q-)MC samples used. We tried 32, 64, 128, and 256 samples and discovered that the performance achieved was almost indistinguishable, so we decided to use 128 samples in the numerical experiments presented in this paper, as discussed in Section 4.1 of the supplement. We observed this robustness for problems with both a small (drop-wave) and medium (epidemic model calibration) number of nodes, so the function network structure does not seem to have an effect on this. We believe this robustness is in great part due to the use of quasi-Monte Carlo samples instead of independent Monte Carlo samples since the spread of the former adapts to the number of samples used.
>
> $\textbf{Q4:}$ There is no discussion of observation noise... I expect that the empirical performance would remain the same... It would be good to discuss and demonstrate performance under noisy observations.
>
> $\textbf{A4:}$ This is a fair point. As suggested, noisy evaluations can be easily handled by our statistical model. Conceptually, our acquisition function can also be generalized to handle noisy evaluations following an analogous approach to the one used by Letham et al. (2019). The resulting acquisition function would be more computationally expensive but we think it would still be broadly applicable. A less principled solution that is similar to ones used in practice in other settings is to replace $f_n^* = \max_{i=1,\ldots, n}f(x_i)$ by $f_n^* = \max_{i=1,\ldots, n}\mathbb{E}_n[f(x_i)]$. This method has virtually the same computational cost as our current acquisition function. Motivated by your question, we implemented this method and found that it delivers outstanding performance compared to noise-aware versions of the standard BO methods. For the Alpine2 problem, for example, the mean objective function value of this method after 100 evaluations is 396.3, whereas the mean objective function of noisy-EI is 137.82  (we are maximizing, so bigger is better); and for the epidemic model calibration problem, the mean log regret of this method after 50 evaluations is -7.22, whereas the mean log regret of noisy-EI after 150 evaluations is -0.73 (here, smaller is better). In the former problem, we set the noise level (i.e., noise standard deviation) of each node equal to 1.0, and in the latter one, we set it equal to 0.1. We also plan to implement the more principled generalization discussed above and assess its performance; we expect it to perform even better. We will discuss the extension of our work to the noisy setting in the revised version of the paper.
>
> $\textbf{Q5:}$ It would be interesting to see a head-to-head comparison with deep GPs. How different are the runtimes (including inference times) and empirical performances?
>
> $\textbf{A5:}$ Thank you for your suggestion. We will compare against a deep GP model using the true underlying network architecture but without access to the intermediate node evaluations. We expect EI-FN to substantially outperform this method because it has access to more information. Moreover, in contrast with our model, inference for this deep GP model is no longer exact and requires computationally intensive inference and marginalization over the unobserved output of intermediate nodes, considerably impacting runtime (if inference is done precisely) or performance (if substantial inference errors are tolerated) or both.
>
> $\textbf{Q6:}$ Since the core contribution is modeling each node in the function network with a distinct GP, it would be good to see more evaluation of the function network model's predictive performance compared to alternative modeling choices…
>
> $\textbf{A6:}$ Thank you for your excellent suggestion. We will include a study of accuracy in the final version of the paper, showing MSE under our statistical model, deep GPs, and vanilla GPs. We believe that our method will outperform both benchmarks as it has access to more information. While our focus remains on optimization and view optimization performance as the most important, this accuracy study will help to illustrate why our method outperforms the SOTA.

---

> ### Author Response · Authors · 2021-08-21
> **Response follow-up**
>
> Dear reviewer vzc8,
>
>
> We would be very grateful if you could confirm whether our response has addressed your concerns, and let us know if any issues remain. Below we recap the main takeaways from our response:
>
>
> $\textbf{A1}$ addresses your concern about the strength of the theoretical guarantee and explains the insight it provides in light of Proposition 2.
>
> $\textbf{A2}$ addresses your concern about the technical novelty of our work and clarifies that it lies in our function network model, which views a class of important problems in a new way that enables dramatic performance gains over the previous state of the art.
>
> $\textbf{A3}$ discusses the effect of the number of (q-)MC samples used in our acquisition function.
>
> $\textbf{A4}$ discusses two simple extensions of our approach to handling noisy evaluations, and presents results for one of these extensions in two of our numerical experiments, which suggest that our approach delivers dramatic performance gains under noisy evaluations as well.
>
> $\textbf{A5}$ discusses a new benchmark method that we will include in our empirical evaluation based on your suggestions, and explains why we expect it to underperform our approach both in terms of computational cost and performance.
>
> $\textbf{A6}$ discusses a study we will conduct to assess the predictive performance of various predictive models to provide additional insight into substantial performance gains achieved by our approach.
>
>
> Thank you again for your valuable feedback. We look forward to answering any other questions you may have.
>
>
> Sincerely,
>
> The authors

---

> > ### Comment · Reviewer_vzc8 · 2021-09-02
> > **Re: Response follow-up**
> >
> > Thank you for your responses. You have adequately addressed my questions.
> >
> > Re Q6: In addition to MSE, it would also be good to evaluated how well-calibrated the model uncertainty is, perhaps by looking at test log-likelihood using leave-one-out cross-validation.
> >
> > However, the primary reason for my score is that the sole methodological contribution of the paper is demonstrating that choosing to model known network structure and leveraging intermediate observations (at network nodes) improves optimization performance. This is a simply a wise modeling choice for a practitioner applying BO. Given the model (a set of distinct GPs each modeling a node in the network), the proposed method simply uses MC-based EI. The use of the SAA for optimizing MC acquisition functions is studied by Balandat et al, 2020, and sampling from the proposed network model is straightforward. Any domain knowledge should be accounted for by a practitioner, especially if there are intermediate observations and known network structure that can aid in the modeling. I see this as "doing due diligence" rather than a significant contribution. Put another way, if presented with an applied problem with known network structure and intermediate observations at each node, it would be ill-advised to ignore that structure and additional information, and the proposed approach is the obvious, simplest modeling choice to exploit this structure that works nearly "off-the-shelf".

---

> > > ### Author Response · Authors · 2021-09-02
> > > **Thank you for your confirmation and a discussion on the significance of our contribution**
> > >
> > > Dear reviewer vcz8,
> > >
> > > We would like to start by thanking you for confirming that our response has adequately addressed your questions. We also want to thank you for your latest suggestion on how to assess how well-calibrated our model’s uncertainty is.
> > >
> > > With respect to the significance of our contribution, we summarize below why our paper is significant and merits publication at NeurIPS 2021.
> > >
> > > As noted by you and all the other reviewers, our approach delivers remarkable performance gains. We agree that, if one knew that leveraging network structure was a path toward dramatic performance improvements, our approach to doing so is a natural one. However, we believe it is not obvious that such dramatic performance gains are possible by modeling the individual nodes of the network. In fact, we think this is the reason why such an approach has not been discussed or applied in the literature so far. When a method is known to deliver powerful performance improvements, it is often adopted quickly by practitioners and rapidly becomes widely known.
> > >
> > > Our paper, which provides an extensive empirical evaluation demonstrating this remarkable phenomenon along with a theoretical result shedding light on why it occurs, could thus play the important role of informing the general ML community about this huge area of opportunity and how to take advantage of it. Given this, we believe the simplicity of our approach is a strength rather than a weakness, as this will allow practitioners to easily adopt it and thus save, with little effort, valuable computational and economic resources when applying BO to this important class of problems.
> > >
> > > Sincerely,
> > >
> > > The authors

---

### Official Review · Reviewer_LhdJ · 2021-07-17

**Rating:** 7
**Confidence:** 4

**Summary:**

This paper presents an extension of Bayesian optimization with Gaussian processes and expected improvement for target functions that are the results of composite functions from a DAG model. The paper introduces an extension of the expected improvement for the specific case which does not allow for a closed form solution. Instead the authors approximated the acquisition function with a sample approximation and optimized it using sample average approximation. The paper also includes some theoretical results in terms of consistency (although not convergence) and extensive results.

**Limitations And Societal Impact:**

Yes

**Main Review:**

The paper extends on the work of Astudillo and Frazier, 2019 for general compositions of functions. In fact, even the structure of the paper is similar which simplifies the reading process if you are familiar with the previous paper. However, I think that this paper abuses the supplementary material. The paper should be self contained, but the text includes 14 times the word “supplement”. I understand that space constraints are an issue, especially after extensions from previous revisions. For example, in my opinion it would be better to fully describe and explain a couple of experiments in the main text, even including Figure 2 as is, instead of half describing all of them.

The extension and the method proposed seem original and, as pointed out in the related work, the problem can be motivated in many areas of science and engineering. I think the NeurIPS community might be quite interested in the RL setup.

The whole point of the statistical model and the SAA optimization is the result of the posterior distribution not being a Gaussian in the general case. However, the samples of any output h are drawn from Gaussian distributions, including hK = g. Furthermore, the authors recommend using SAA with deterministic samples, because the evaluations of the acquisition function are noisy. However, isn’t that the result of using a naive MC strategy? Having a DAG, wouldn’t be better to use belief propagation or exploit the structure with some MCMC or particle filter and have a more accurate representation of the acquisition function? I can see how BP can be much more expensive than the suggested approach, but this method is already quite expensive compared to regular BO. Furthermore, it seems that the use of SAA is not clearly motivated, being the default method for MC-based acquisitions in BoTorch.

The results are quite interesting. Clearly, the method reaches SOTA, even compared to simpler versions, such as EI-CF. However, the authors mix absolute value with log(regret). For instance, they claim that the improvement in the robot problem is quite large. However, any real robot application can be considered “solved” with log(regret) = -2 or -3. From a pure numerical/benchmark point of view, the result is still interesting, but then I would like to have a more realistic RL setup. For example, the suggested setup or learning a sequence of waypoints as a policy has been previously done in BO for robot navigation:

Ruben Martinez-Cantin, Nando de Freitas, Eric Brochu, Jose Castellanos and Arnaud Doucet (2009) A Bayesian Exploration-Exploitation Approach for Optimal Online Sensing and Planning with a Visually Guided Mobile Robot. Autonomous Robots - Special Issue on Robot Learning, Part B, 27(3):93-103.

Small comments:

-Given that KG is sometimes superior to EI, would it be possible to have KG-FN or is there a limitation that could prevent it?

-If the robot pushing and epidemic model are in the perfect setup for EI-CF, how would you explain the improvement of EI-FN?

----
Notes after author response: The authors have addressed most of my concerns. I'm glad that the new experiment worked so well. It clearly shows the advantages of the proposed method and I have updated my score as a result.

As a final suggestion, one of the advantages of the method is clearly computational cost. Thus, I would address that with some results or comments regarding that. The fact that "vanilla" KG is x4 more expensive than EI-FN is remarkable, as one would expect EI-FN to be the more expensive given the number of GPs to manage.


**Time Spent Reviewing:**

8

---

> ### Author Response · Authors · 2021-08-11
> **Author response to reviewer LhdJ**
>
> We would like to thank you for your feedback and questions. We are glad that you found the results of our paper “quite interesting”. Below, we respond to your questions and comments. We look forward to answering any other questions you may have.
>
> $\textbf{Q1:}$ If the robot pushing and epidemic model are in the perfect setup for EI-CF, how would you explain the improvement of EI-FN?
>
> $\textbf{A1:}$  While these problems have a composite structure that allows EI-CF to dramatically outperform standard BO, they have additional network structure that EI-CF cannot exploit and that EI-FN can. Since EI-FN leverages the composite structure of the objective as part of the function network structure, the information leveraged by EI-FN is thus strictly greater than the one leveraged by EI-CF. This explains why EI-FN outperforms EI-CF.
>
> $\textbf{Q2:}$ ...it would be better to fully describe and explain a couple of experiments in the main text... instead of half describing all of them.
>
> $\textbf{A2:}$ Thank you for the suggestion. We will do this in the revised version of our paper.
>
> $\textbf{Q3:}$ ...any real robot application can be considered “solved” with log(regret) = -2 or -3... From a pure numerical/benchmark point of view, the result is still interesting, but then I would like to have a more realistic RL setup.
>
> $\textbf{A3:}$ We will acknowledge this in the paper when discussing the results from robot pushing which, as the reviewer points out, is interesting when viewed as a numerical benchmark. Indeed, this problem has been used as a benchmark in a variety of other BO papers, which was part of our motivation for including it. At your request, we also plan to add an additional RL test problem to more fully demonstrate our approach in the RL setting.
>
> $\textbf{Q4:}$...the authors recommend using SAA with deterministic samples... isn’t that the result of using a naive MC strategy? ...wouldn’t be better to use belief propagation or exploit the structure with some MCMC or particle filter and have a more accurate representation of the acquisition function?  I can see how BP can be much more expensive...
>
> $\textbf{A4:}$ Belief propagation (BP) is not easy to apply in our setting because each node in our model is an infinite-dimensional random field. While there is a small amount of work extending BP to such settings (see Seeger, 2007, “Gaussian process belief propagation”), we are not aware of computationally efficient practical methods accomplishing this.  Moreover, one would still need to integrate the posterior distribution obtained from such an infinite-dimensional generalization of BP to calculate the resulting expected improvement.  MCMC, particle filtering, and other methods for drawing samples from the posterior would not seem to offer benefit because we are already able to efficiently draw samples from the exact posterior distribution. Our approach combining exact sampling and SAA is able to solve the acquisition function optimization problem in a practically relevant amount of time. We also considered stochastic gradient ascent, following an analogous approach to the one proposed by Astudillo & Frazier (2019), and found in preliminary experiments that it was substantially less efficient than SAA.
>
> $\textbf{Q5:}$ ...would it be possible to have KG-FN or is there a limitation that could prevent it?
>
> $\textbf{A5:}$ Conceptually, one can extend KG to the function networks setting using our predictive model following an analogous approach to the one that we pursue for EI. However, standard KG is already fairly computationally expensive. For example, in the epidemic model calibration and robot pushing test problems, standard KG is approximately 4 times more expensive than our method despite using a vanilla (single-output) GP model. This occurs because, due to its non-myopic definition, KG requires special optimization routines such as stochastic gradient ascent or “one-shot optimization” (see, e.g., Balandat et al. 2019). Implementing these in the context of our statistical model in a naive fashion would make KG-FN extremely computationally expensive. However, extending KG and other more sophisticated acquisition functions to our setting in a computationally tractable way (potentially by introducing approximations) is definitely an interesting direction for future research.

---

> ### Author Response · Authors · 2021-09-01
> **New RL problem**
>
> Dear reviewer LhdJ,
>
> We would like to thank you again for your valuable feedback, including your suggestion to add experiments demonstrating the power of our method in more realistic RL environments:
>
> ​”The results are quite interesting. Clearly, the method reaches SOTA, even compared to simpler versions, such as EI-CF. However, the authors mix absolute value with log(regret). For instance, they claim that the improvement in the robot problem is quite large. However, any real robot application can be considered “solved” with log(regret) = -2 or -3. From a pure numerical/benchmark point of view, the result is still interesting, but then I would like to have a more realistic RL setup. For example, the suggested setup or learning a sequence of waypoints as a policy has been previously done in BO for robot navigation:
>
> Ruben Martinez-Cantin, Nando de Freitas, Eric Brochu, Jose Castellanos and Arnaud Doucet (2009) A Bayesian Exploration-Exploitation Approach for Optimal Online Sensing and Planning with a Visually Guided Mobile Robot. Autonomous Robots - Special Issue on Robot Learning, Part B, 27(3):93-103.”
>
> As you observed, our framework is well-suited to learning a sequence of waypoints for robot path planning, such as the one considered by Martinez-Cantin et al. 2009. We took your suggestion and applied our method to a similar problem, which we describe below along with the results obtained. We will include this new experiment in the revised version of our work, along with another RL-based experiment that is currently under preparation (resulting thus in a grand total of 10 test problems).
>
> This problem is obtained by adapting the Fetch environment (see Plappert et al. 2018. "Multi-goal reinforcement learning: Challenging robotics environments and request for research") from OpenAI Gym. The goal is to move the gripper of a robotic arm to a target location with only three movements. We formulated this problem as a function network with 3 nodes, each representing a movement of the robotic arm. Each of these nodes takes as input the current location of the gripper and a vector of forces to be applied to the robot arm in that step, and produces as output the location of the arm after this movement is complete. (Note that the output of each node is 3-dimensional and thus this can also be thought of as a function network with 9 (single-output) nodes). The objective to minimize is the distance between the gripper and the target in the final step.
>
> We compare our method (EI-FN) with the EI-CF method of Astudillo & Frazier 2019, EI (which is the method used by Martinez-Cantin et al. 2009 for waypoint planning), KG, and random search. The results are summarized in the table below and show the average best objective value found ($\pm$ 1.96 times the standard error) after a given number of evaluations for each of the methods (note that this objective value instead of log(regret)).
>
>
> |Method / Number of evaluations | 10                    | 20 |                       30                                        |
> |-------------------------------------------| :-------: | :---: | :----: |
> |Random           |                            $9.56 \pm 0.69$     |     $8.86 \pm 0.64$     |     $8.50 \pm 0.61$ |
> |KG                      |                         $5.23 \pm 0.79$      |    $1.29 \pm 0.55$      |    $0.96 \pm 0.53$|
> |EI                       |                          $4.13 \pm 0.44$    |      $1.03 \pm 0.17$       |   $0.56 \pm 0.07$|
> |EI-CF                    |                       $1.02 \pm 0.23$      |    $0.42 \pm 0.03$    |      $0.11 \pm 0.04$|
> |EI-FN                    |                       $0.36 \pm 0.01$     |     $0.14 \pm 0.04$      |     $0.03 \pm 0.02$|
>
>
> As in other test problems, we see that EI-FN substantially outperforms all the other methods, including EI-CF, which also outperforms standard BO methods significantly. In particular, we note that, with only 10 evaluations, EI-FN achieves a better average objective value than EI after 30 evaluations and also that EI-CF after 20 evaluations.
>
> We hope this new experiment more clearly demonstrates the potential of our method to achieve state-of-the-art performance in this class of problems.
>
> Sincerely,
>
> The authors

---

> ### Author Response · Authors · 2021-09-01
> **Thank you for your confirmation and support for our paper**
>
> Dear reviewer  LhdJ,
>
> We sincerely appreciate your confirmation and are very glad that our response improved your opinion of our work. We would also like to thank you for your suggestion on how to alleviate the concerns around the computational cost of our method; we will add such a discussion in the revised version of our paper.
>
> Sincerely,
>
> The authors

---

### Official Review · Reviewer_HdZA · 2021-07-25

**Rating:** 7
**Confidence:** 3

**Summary:**

This paper extends methods for Gaussian process "Bayesian Optimization" (BO)  of vectors of hyperparameters to structural networks of function, to include hyperparameters from functions of intermediate stages. Unlike a conventional approach, this method considers the output of intermediate stages.  The hypothesis addressed by the paper that including additional information from intermediate stages is a reasonable one, however it raises a challenge since despite the intermediate functions each being modelled as a GP, the final output g is not necessarily Gaussian.  To accommodate this, they apply sample average approximation to sample from the posterior on the final output.

The authors apply this method in the context of several application areas, having run experiments on 8 problems with results showing substantially faster convergence than competing methods.

This paper offers a convincing empirical study together with theoretical convergence results. This is an interesting paper with useful learning to the general community.

**Ethical Concerns:**

No ethical concerns arise in this current work.

**Limitations And Societal Impact:**

The authors are cognizant that this technique can be used for ends that are both socially valuable or destructive, and state as much.  In it's current nascent form there is no reason to presume it's use would encourage abusive uses.

**Main Review:**

The paper's approach is to provide an approximation to the posterior for the acceptance function, EI-FN, using a sampling method based on the 'so-called re-parameterization trick."  In Proposition 1 they show a convergence proof for the function.   The optimization efficiency they claim is demonstrated in the set of empirical problems on both simulated and actual data.

As for the evaluation criterion used, in the calculation of the approximated EI-FN function, the method generates M samples (even from the posterior distribution without actually evaluating the network) that can potentially reduce the variance of this evaluation. It appears that the posterior distribution is still estimated only using the n samples. When counting the number of iterations,  in Figure 2, is the x-axis  only counting n as the iteration number?  One wonders if the the total number of function evaluations would be a fairer basis for comparison , given the implication of this paper is that this “synthetic” M samples are the key to improve the convergence rate. This leads to a speculation if there exists other ways of generating the samples that can possibly lead to even better convergence rate (either based on posterior distribution or not).

As claimed, basic information-theoretic notions lead one to believe that the intermediate values used in the EI-FN function should lead to more efficient sampling estimates, hence faster optimization.  This property, although demonstrated in the empirical results, is not supported, and deserves further study since the analysis provided only hints in broad terms why it comes about. Convergence rates alone are only an indirect explanation for the observed efficiency.  Perhaps there are diagnostics derived from comparing conventional acquisition functions with the approximations generated?

To speculate further, the example network structures resemble probabilistic graphical models, so one could imagine an inferential approach to compute distributions over the x's that would enable estimation of how much improvement is due to an x's additional information.   This is not to criticize the work to date, but it's surprising that thoughts about this did not get a mention.

**Time Spent Reviewing:**

3 hours

---

> ### Author Response · Authors · 2021-08-11
> **Author response to reviewer HdZa**
>
> We sincerely appreciate your feedback and questions. We are glad that you found our paper “interesting... with useful learning to the general community”, with results that “are quite interesting” and “a convincing empirical study together with theoretical convergence results.” We respond to your comments and questions below, and look forward to answering any other questions you may have.
>
> $\textbf{Q1:}$ When counting the number of iterations, in Figure 2, is the x-axis only counting n as the iteration number? One wonders if the total number of function evaluations would be a fairer basis for comparison, given the implication of this paper is that these “synthetic” M samples are the key to improve the convergence rate.
>
> $\textbf{A1:}$ Yes, the x-axis in Figure 2 shows the number of function evaluations $n$; i.e., the number of times that the entire function network has been evaluated in reality. These evaluations are what takes a long time (often hours in the applications we are interested in) and so we track them as the primary means of measuring performance. The “synthetic” samples are generated extremely fast (milliseconds) and so it is more meaningful to focus on $n$. Section 4 of the supplement shows the wall-clock time required to optimize the acquisition function, which includes the computation time required to generate the synthetic samples.
>
> $\textbf{Q2:}$ As claimed, basic information-theoretic notions lead one to believe that the intermediate values used in the EI-FN function should lead to more efficient sampling estimates, hence faster optimization. This property, although demonstrated in the empirical results, is not supported, and deserves further study since the analysis provided only hints in broad terms why it comes about.
>
> $\textbf{A2:}$ Thank you for this suggestion! Indeed, it is possible to shed light on the great empirical performance of our method following an information-theoretic perspective. We will add a proposition showing that the variance and entropy of the posterior on the objective function’s value at a given point under our model are lower than under a deep GP with the same architecture. This occurs because the first posterior is obtained from the second by conditioning on additional data (the observations of intermediate nodes).
>
> $\textbf{Q3:}$ To speculate further, the example network structures resemble probabilistic graphical models, so one could imagine an inferential approach to compute distributions over the x's that would enable estimation of how much improvement is due to an x's additional information. This is not to criticize the work to date, but it's surprising that thoughts about this did not get a mention.
>
> $\textbf{A3:}$ We will discuss the relation to probabilistic graphical models in the revised version of our paper --- our model is a probabilistic graphical model, but where nodes represent infinite-dimensional random fields rather than finite-dimensional random variables or vectors (see, e.g., Seeger. 2007 “Gaussian process belief propagation”) more common in the literature. Our approach performs inference using exact sampling and uses the resulting sampling-based approximation to the posterior distribution to compute the improvement from an x’s additional information. We will discuss possibilities for other inference methods in the final version of the paper, including belief propagation for random fields (Seeger, 2007). However, our results demonstrate that our exact sampling + SAA approach is practical for this task and is likely much more efficient than this alternate approach; but we agree it is possible that future work may develop even more computationally efficient methods.

---

> ### Author Response · Authors · 2021-09-02
> **Thank you for your valuable feedback and support for our paper**
>
> Dear reviewer HdZA,
>
> We would like to thank you again for your valuable feedback. We are glad that you saw the value in our work and we particularly appreciate your suggestion to explore information-theoretic explanations of the improvements provided by our approach --- this will be a useful addition to the paper.
>
> With the end of the discussion phase quickly approaching, please let us know if you have any additional comments or questions after reading our response.
>
> Sincerely,
>
> The authors

---

### Official Review · Reviewer_r59T · 2021-07-28

**Rating:** 6
**Confidence:** 3

**Summary:**

This work investigates the problem of Bayesian optimization of the output of a network of functions. The network is organized as a directed acyclic graph, where each function takes as input the output of its parent nodes as well as (part of ) the input. Contrary to the standard setting that only observes the final output, this work assumes that intermediate nodes can also produce an output (which can be stochastic or deterministic). The authors model the nodes of the network using Gaussian processes, and employ an acquisition function that corresponds to the expected improvement with respect to the implied posterior on the final output under a proper statistical model. The authors optimize the acquisition function using sample average approximation. The authors show that their approach satisfies asymptotic convergence. Various experiments on synthetic and real problems demonstrate that the improved model and acquisition function result in visibly improved solution quality, at the expense of the runtime cost.

**Limitations And Societal Impact:**

Limitations and potential societal impact are discussion in the Conclusion.

**Main Review:**

Strengths
- This work introduces a new setting for Bayesian Optimization (BO) that is highly relevant in practice, e.g., in engineering applications. This work seems to be the first to address this setting. For this reason, this work is fairly significant.
- On the theory front, the work includes a convergence proof. Furthermore, it investigates the interesting property that the acquisition function for function networks does not need to measure the optimization domain densely. This suggests a possible mechanism via which the new acquisition function can lead to more efficient BO.
- The paper is well written and easy to follow.
- The experimental section is well written and demonstrates that the new model and acquisition function achieve superior result in terms of the objective value. The problems considered in the experimental study are diverse and quite realistic.

Concerns
- Using GPs to model the various nodes in the function network, and optimizing the acquisition function via sample average approximation are not particularly novel for that setting. Even though the problem statement in this work is novel, the authors have chosen a rather straightforward approach to tackle it. That being said, they do show consistency.
- Convergence rates are not discussed, only consistency.
- The method is quite time-consuming compared to simpler competitors, as illustrated in the runtimes. This problem can be exacerbated for large function networks with several nodes.

Additional questions to authors
- How does the computational complexity depend on the number of nodes in the function network? Experimentally, function networks of up to how many nodes have the authors been able to handle?
- The framework can deal with both deterministic and non-deterministic nodes. What about nodes whose output is hidden? Can the framework handle such cases for an arbitrary node (assuming of course the DAG structure for the function network)?

**Time Spent Reviewing:**

4

---

> ### Author Response · Authors · 2021-08-11
> **Author response to reviewer r59T**
>
> We would like to thank you for your feedback and questions. We are glad that you found our paper “well written and easy to follow”, with a problem setting “that is highly relevant in practice”, and an empirical evaluation that “demonstrates that the new model and acquisition function achieve superior results”.  Below, we respond to the questions and comments raised in this review. We look forward to answering any other questions you may have.
>
> $\textbf{Q1:}$ How does the computational complexity depend on the number of nodes in the function network? Experimentally, function networks of up to how many nodes have the authors been able to handle?
>
> $\textbf{A1:}$ The computational complexity of our method grows linearly with the number of nodes. We have been able to handle up to 9 nodes on a personal laptop. We believe that our method could be scaled to a few tens of nodes (~30-50) using a more powerful computer, but it does not yet scale to hundreds of nodes, as discussed in the conclusion. We are confident that our method is still broadly applicable even without scaling beyond ~50 nodes. This is particularly true because nodes can be collapsed together to produce a smaller network. In many problems, we expect that this smaller sub-network will still provide a significant benefit over methods that do not leverage the network structure. At the same time, scaling our approach to a larger number of nodes is an exciting direction for future research.
>
> $\textbf{Q2:}$ Even though the problem statement in this work is novel, the authors have chosen a rather straightforward approach to tackle it.
>
> $\textbf{A2:}$ We agree that the technical tools required to compute and optimize our acquisition function are simple despite also providing a large improvement over the state-of-the-art. We see this as an advantage of our approach. Fundamentally, our core contribution is one of modeling --- of pointing out that orders of magnitude in improvement over the state of the art are possible with an efficient method, showing that reliable and trusted ideas can be combined and brought to bear on a novel model to produce great benefit. We also hope that the simplicity of our approach will make it easier to be adopted by practitioners and thus have a greater impact.
>
> $\textbf{Q3:}$ Convergence rates are not discussed, only consistency.
>
> $\textbf{A3:}$ For adaptive methods like ours in which potentially complex behavior supports good practical performance, convergence rates are difficult to show.  Indeed, convergence rates are unknown for many important and well-studied acquisition functions, such as predictive entropy search and the knowledge-gradient method. As with these acquisition functions, proving convergence rates in our setting is quite challenging because EI-FN lacks a closed-form analytical expression amenable to be analyzed. Moreover, our consistency result provides additional insight into why our method works so well by showing that consistency is possible without measuring densely, allowing the exclusion of entire parts of the search space. Nevertheless, establishing convergence rates is an exciting direction for future work.
>
> $\textbf{Q4:}$ The method is quite time-consuming compared to simpler competitors, as illustrated in the runtimes.
>
> $\textbf{A4:}$ While our method is more time-consuming compared to other simpler BO methods, the performance gains it provides more than make up for the additional computation required. Consider, for example, the epidemic model calibration problem, and suppose that each evaluation of the epidemic model takes 30 minutes (recall that BO is typically applied in problems where evaluations are time-consuming). Computing the point suggested by our method takes approximately 5 minutes on average, as can be seen from Table 1 in the supplement. However, with only ~10 evaluations, our method is able to achieve the same level of accuracy achieved by standard BO methods with 150 evaluations. Translated into time, this means that in 5 * 10 + 30 * 10 = 350 minutes = 5.8 hours, our method achieves the same level of accuracy as standard BO methods in 150 * 30 = 4500 minutes = 3.1 days.  A similar computation for other benchmarks reveals that EI-FN typically saves substantial wall-clock time (acquisition time plus objective function evaluation) as long as the objective function evaluations are reasonably time-consuming.
>
> We would also like to emphasize that acquisition times on the order minutes are not unusual for modern BO methods, such as KG. In fact, for the epidemic model calibration and robot pushing test problems, our method is approximately 4 times faster than KG. (Despite using a vanilla GP model, KG is slow because it requires special optimization techniques due to its non-myopic definition).
>
> $\textbf{Q5:}$ What about nodes whose output is hidden? Can the framework handle such cases for an arbitrary node...?
>
> $\textbf{A5:}$ It is possible, but inference for our statistical model would no longer be exact and would require approximations typically used for deep GPs.

---

> > ### Comment · Reviewer_r59T · 2021-09-05
> > **read author response**
> >
> > I thank the authors for their response. The rebuttal has addressed my concerns, and I encourage the authors to include these clarifications in the next version of their manuscript.

---

> > > ### Author Response · Authors · 2021-09-09
> > > **Thank you for your confirmation**
> > >
> > > Dear reviewer r59T,
> > >
> > > Thank you for confirming that our response has adequately addressed your concerns. We will make sure to take into account all the suggestions and concerns raised by the reviewing team in the revised version of our paper.
> > >
> > > Sincerely,
> > >
> > > The authors

---

> ### Author Response · Authors · 2021-09-02
> **Confirmation and thank you for your support**
>
> Dear reviewer r59t,
>
> We would like to start by thanking you again for your valuable feedback and support for our paper.
>
> With the end of the discussion phase quickly approaching, we would be very grateful if you could confirm whether our response has adequately addressed your concerns. Please do not hesitate in letting us know if any other questions remain.
>
> If our response or comments made by other reviewers have improved your opinion of our work, we would kindly ask you to consider increasing your rating a bit as this would significantly improve the chances of our paper being accepted.
>
> Thank you so much in advance for your consideration.
>
> Sincerely,
>
> The authors

---

### Decision · Program_Chairs · 2021-09-27

**Decision:**

Accept (Poster)

**Comment:**

This paper addresses Bayesian optimization which leverages the intermediate outputs as well as final outputs when they are available by evaluations in the function network. The problem setting is novel, extending the previous work [Astudillo, R. and Frazier, P., 2019]. A cascade of GPs is used as a surrogate model, leading to non-Gaussian process. A sample average approximation approach is used to optimize the EI. Strong empirical results are provided to demonstrate that the proposed method indeed achieves superior performance. The strength of this paper is in the novel problem setting, its usefulness in engineering applications mainly for practitioners. The downside is in the limited novelty in the proposed solution, since it is a direct employment of GPs and SAA. During the committee discussion period, I had a few communications with reviewers. The problem setting, which appears often in engineering applications, is valuable for further studying.  Thus, even if the proposed solution is not novel, this work studies an important and valuable problem, and could serve as a starting point for new works that could develop more sophisticated algorithms or deeper theory for the DAG-dependency setting.